# BioTamperNet: Affinity-Guided State-Space Model Detecting Tampered Biomedical Images

**Soumyaroop Nandi**[1,2], **Prem Natarajan**[1,2,3]

[1]USC Information Sciences Institute, Marina del Rey, CA, USA
[2]USC Thomas Lord Department of Computer Science, Los Angeles, CA, USA, [3]Capital One
{soumyarn,premkumn}@usc.edu

## Abstract

We propose BioTamperNet, a novel framework for detecting duplicated regions in tampered biomedical images, leveraging affinity-guided attention inspired by State Space Model (SSM) approximations. Existing forensic models, primarily trained on natural images, often underperform on biomedical data where subtle manipulations can compromise experimental validity. To address this, BioTamperNet introduces an affinity-guided self-attention module to capture intra-image similarities and an affinity-guided cross-attention module to model cross-image correspondences. Our design integrates lightweight SSM-inspired linear attention mechanisms to enable efficient, fine-grained localization. Trained end-to-end, BioTamperNet simultaneously identifies tampered regions and their source counterparts. Extensive experiments on the benchmark bio-forensic datasets demonstrate significant improvements over competitive baselines in accurately detecting duplicated regions. Code - https://github.com/SoumyaroopNandi/BioTamperNet

## 1 Introduction

Scientific integrity is fundamental to credible research, yet instances of image manipulation in biomedical publications continue to raise serious concerns. Such misconduct not only damages scientific reputation but also contributes to the reproducibility crisis and imposes significant financial costs on the research community Miyakawa (2020); Bik et al. (2018); Stern et al. (2014). Manual review processes are often inadequate, particularly when manipulations involve subtle duplications or complex alterations Bik et al. (2016). As image forgeries grow increasingly sophisticated, there is a critical need for automated forensic tools to ensure the reliability and integrity of scientific findings.

Biomedical images exhibit significant semantic and visual diversity, posing greater challenges than in natural images. They vary widely in structure, content, and acquisition techniques. Following Sabir et al. (2021), we categorize biomedical images into four types: *microscopy* (cell and tissue images under a microscope), *blot/gel* (protein, RNA, and DNA analysis such as Western blots), Fluorescence-Activated Cell Sorting-*FACS* (scatter plots for cytometry analysis), and *macroscopy* (miscellaneous biomedical images such as scans and leaves), as shown in Figure 1. Each modality introduces distinct forensic challenges, motivating the need for models adaptive across diverse image types.

The BioFors dataset Sabir et al. (2021) defines three key forensic tasks, illustrated in Figure 2. *External Duplication Detection (EDD)* involves identifying duplicated regions between two images, arising from either image repurposing (cropping from a shared source) or splicing (pasting regions across images). *Internal Duplication Detection (IDD)* focuses on detecting repeated regions within a single image without assuming the manipulation type; often, it is unclear which duplicated patch, if any, is authentic. *Cut Sharp Transition Detection (CSTD)* targets sharp transitions indicative of stitched or composite images, unique to the biomedical domain. Additionally, blot/gel images exhibit *patch removal*, where regions are blurred or inpainted to obscure unfavorable results.

In this work, we propose BioTamperNet, a unified framework based on State Space Models (SSMs) for detecting duplicated and manipulated regions in biomedical images. BioTamperNet consists of two key components: an *affinity-guided self-attention module* to detect similar regions within an image (for internal duplications), and an *affinity-guided cross-attention module* to identify duplicated

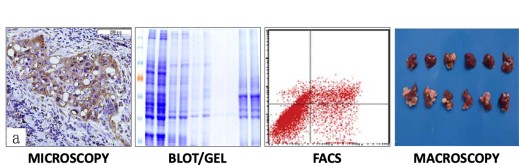

Figure 1: Microscopy, Blot/Gel, FACS, Macroscopy in BioFors Sabir et al. (2021).

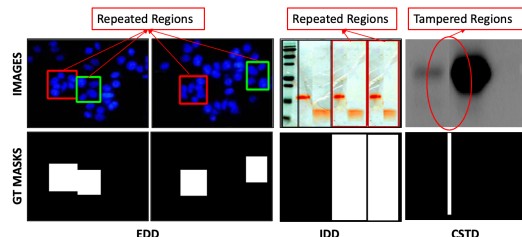

Figure 2: EDD uses image pairs for detection; IDD and CSTD use single image each.

regions between image pairs (for external duplications). By leveraging SSMs as efficient attention mechanisms, BioTamperNet accurately localizes duplications while maintaining efficiency.

As the BioFors dataset Sabir et al. (2021) lacks manipulated training samples, we create synthetic datasets by inserting duplicated patches with extensive augmentations (scaling, rotation, cropping, flipping, noise addition), and further enhance realism using GAN-generated patches and blending techniques. To address EDD, BioTamperNet operates on image pairs, predicting source and target duplicated regions. For IDD and CSTD, we synthetically generate image pairs by cutting forged single images, allowing a single architecture to handle both tasks without requiring modifications. This unified design simplifies training and inference across diverse biomedical manipulations.

Our contributions are summarized as follows:

- We propose BioTamperNet, an unified architecture for both paired-image and single-image forgery localization in biomedical images.
- We design affinity-guided self-attention and cross-attention modules based on lightweight SSM-inspired linear attention for structured duplication detection.
- We generate synthetic and GAN-augmentation, mitigating scarcity in biomedical forensics.
- We achieve state-of-the-art performance on the BioFors Sabir et al. (2021) test set, which contains real-world forged experimental images from retracted scientific publications.

## 2  TRAINING DATASET

BioFors Sabir et al. (2021) provides a benchmark for biomedical image forgery detection but its train split only includes pristine images without forgeries or corresponding ground truth masks. To enable supervised training of BioTamperNet, we generate synthetic datasets with paired images and ground truth masks from the train split of Sabir et al. (2021). The training setup for EDD differs from that of IDD and CSTD, as EDD requires a pair of images with corresponding ground truth masks, whereas IDD and CSTD training use a single image with its ground truth mask. To unify training within a single model, we devised a novel strategy for BioTamperNet. BioTamperNet is designed to handle EDD by processing image pairs and predicting a pair of manipulated masks as illustrated in Figure 3. To incorporate IDD and CSTD images into the EDD training framework, we split each IDD and CSTD image into two parts, generating a corresponding pseudo-pair of input images and ground truth masks, enabling their use within the EDD training setup, as illustrated in Figure 4.

To increase data diversity and realism, we apply extensive augmentations, including geometric transformations (scaling, rotation, flipping, cropping) and noise perturbations. Additionally, we employ Generative Adversarial Networks (GANs) to synthesize realistic duplicated patches that better simulate complex manipulations found in real biomedical images. This synthetic, augmented, and blended training strategy improves BioTamperNet's robustness, generalization, and adaptability to diverse forgery patterns observed in biomedical research publications.

## 3  PROPOSED METHOD

### 3.1  PRELIMINARIES - STATE SPACE MODELS

State Space Models (SSMs) Gu et al. (2021); Gu & Dao (2023); Liu et al. (2024) represent complex sequences using linear time-invariant (LTI) dynamics, generalizing classical approaches such as the

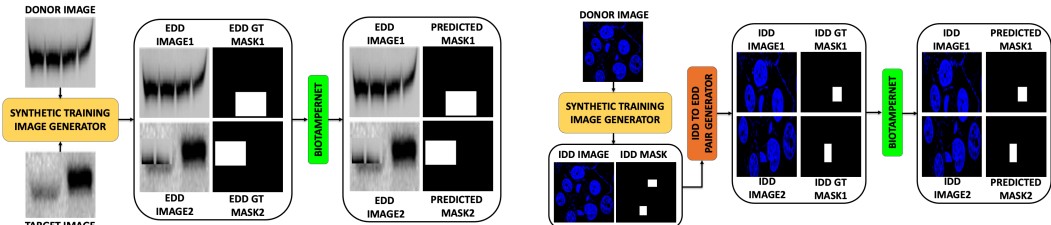

Figure 3: The EDD Synthetic Training Setup.          Figure 4: The IDD Synthetic Training Setup.

Kalman Filter Kalman (1960). Given a continuous input sequence $x(t) \in \mathbb{R}$, the system evolves a hidden state $h(t) \in \mathbb{R}^N$ through learnable dynamics defined by $\mathbf{A} \in \mathbb{R}^{N \times N}$, $\mathbf{B} \in \mathbb{R}^{N \times 1}$, and projects an output via $\mathbf{C} \in \mathbb{R}^{1 \times N}$, trained end-to-end:

$$h'(t) = \mathbf{A}h(t) + \mathbf{B}x(t), \quad y(t) = \mathbf{C}h(t) \tag{1}$$

Discrete-time dynamics incorporate a timescale $\Delta$ using zero-order hold (ZOH) discretization Gu et al. (2021), resulting in the following discrete updates:

$$\bar{\mathbf{A}} = \exp(\Delta\mathbf{A}), \quad \bar{\mathbf{B}} = (\Delta\mathbf{A})^{-1}\left(\exp(\Delta\mathbf{A}) - \mathbf{I}\right)\Delta\mathbf{B}, \quad \bar{\mathbf{C}} = \mathbf{C}$$
$$h_k = \bar{\mathbf{A}}h_{k-1} + \bar{\mathbf{B}}x_k, \quad y_k = \bar{\mathbf{C}}h_k + \bar{\mathbf{D}}x_k \tag{2}$$

Approximating $\bar{\mathbf{B}}$ via Taylor expansion and $\bar{\mathbf{D}}$ working as a residual connection yields:

$$\bar{\mathbf{B}} = (\exp(\mathbf{A}) - \mathbf{I})\mathbf{A}^{-1}\mathbf{B} \approx (\Delta\mathbf{A})(\Delta\mathbf{A})^{-1}\Delta\mathbf{B} = \Delta\mathbf{B}, \quad y_k = \bar{\mathbf{C}}h_k \tag{3}$$

Mamba Gu & Dao (2023) dynamically generates $\mathbf{B}$, $\mathbf{C}$, and $\Delta$ from input $\mathbf{x} \in \mathbb{R}^{L \times D}$ by modeling interactions in long sequences through selective scanning to maintain contextual awareness, where $L$ is the input token size and $D$ is the embedding dimension. VMamba Liu et al. (2024) extends this to 2D via Cross-Scan and Cross-Merge operations, where Visual State Space Blocks process sequences along four directions independently and fuse outputs for global spatial modeling.

### 3.2    BIOTAMPERNET ARCHITECTURE OVERVIEW

BioTamperNet is a deep Siamese architecture for tampered region localization, consisting of a Vision Transformer-based Feature Extractor, a Siamese Duplication Detector with Affinity-guided SSM attention modules, and a lightweight Decoder as illustrated in Figure 5. Given a pair of input images $x_1, x_2 \in \mathbb{R}^{B \times H \times W \times 3}$, with image height $H$ and width $W$, high-dimensional hierarchical feature representations $V_1, V_2 \in \mathbb{R}^{B \times N \times C}$ are extracted using a Vision Transformer pretrained on the four classes of BioFors dataset (classes described in Section 1), where $B$ is the batch size, $N = H \times W$, and $C = 384$ represents the embedding dimension for our proposed Siamese Duplication Detector:

$$V_1 = \text{Feature\_Extractor}(x_1), \quad V_2 = \text{Feature\_Extractor}(x_2) \tag{4}$$

These features are processed by the `Siamese Duplication Detector` module to compute affinity maps, self- and cross-attention features guided by inter-image similarity and duplication cues:

$$V_1', V_2' = \text{Siamese\_Duplication\_Detector}(V_1, V_2) \tag{5}$$

The attention-refined features are decoded into binary tampering mask pairs $O_1 = \text{Decoder}(V_1')$ and $O_2 = \text{Decoder}(V_2')$ via convolutional layers that highlight duplicated regions.

### 3.3    SIAMESE DUPLICATION DETECTOR MODULE

The Siamese Multi-Level Attention (Siamese Duplication Detector) module in Figure 5 is designed to enhance feature representations of paired images by leveraging affinity maps for both self-attention and cross-attention guidance. Given two feature maps $V_1, V_2 \in \mathbb{R}^{B \times N \times C}$, the module returns contextually enriched representations $V_1'$ and $V_2'$ for robust duplicate or tampering detection.

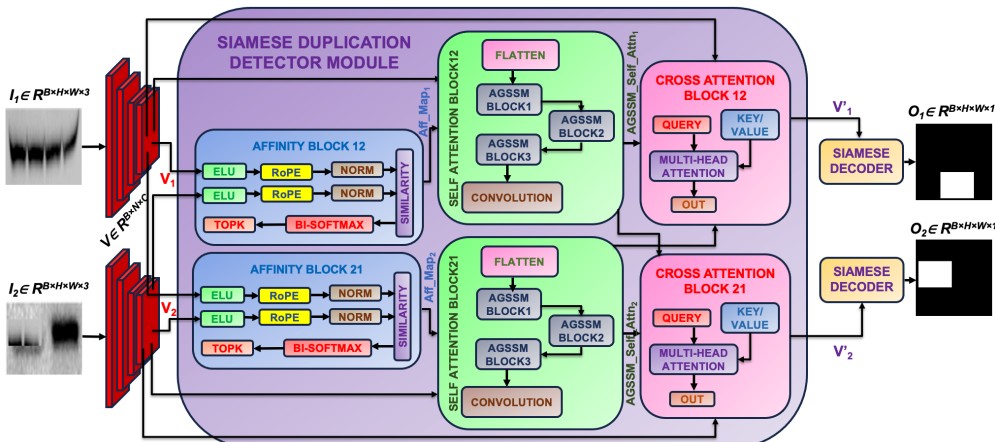

Figure 5: BioTamperNet architecture: Paired input images are processed through feature extraction, affinity-guided self-attention, cross-attention, followed by decoders to produce binary mask pair.

**Affinity Block.** Before computing affinities, we first contextualize spatial tokens using a selective-scan State Space Model (SSM). Given the feature map $V \in \mathbb{R}^{B \times N \times C}$, we apply an input-adaptive SSM backbone implemented using the selective scan operator to obtain SSM-encoded features. The affinity matrix $\text{Aff} \in \mathbb{R}^{B \times N \times N}$ is then constructed over these SSM-encoded representations using the State Space Similarity mechanism introduced in Eq. 2. Specifically, we normalize the term $\bar{\mathbf{C}}h_k$, where $\mathbf{h}_k = \sum_{j=1}^{k} \mathbf{B}_j^\top x_j$ and $\mathbf{n}_k = \sum_{j=1}^{k} \mathbf{B}_j$, following Gu & Dao (2023). The resulting similarity attention is formulated as:

$$y_k = \frac{\bar{\mathbf{C}}h_k}{\bar{\mathbf{C}}n_k} + \bar{\mathbf{D}}v_k, \quad h_k = \bar{\mathbf{A}}h_{k-1} + \bar{\mathbf{B}}v_k \tag{6}$$

Equation 6 defines a global-context similarity mechanism, where the SSM transition matrix $\bar{\mathbf{A}}$ aggregates contextual information from preceding inputs $\mathbf{v}_j$ and outputs $\mathbf{y}_j$ for $j \leq k$. Normalization by $\bar{\mathbf{C}}n_k$ ensures the attention weights sum to one. In practice, this recurrence is implemented using the selective scan operator, and the affinity is computed over SSM-encoded features rather than raw transformer outputs. To stabilize affinity estimation, we apply an ELU activation with a positive shift and inject spatial inductive bias using Rotary Positional Embeddings (RoPE) Su et al. (2024).

$$\bar{\mathbf{C}}_k, \bar{\mathbf{B}}_k = \text{ELU}(\mathbf{V}_k) + 1.0, \quad \bar{\mathbf{C}}_k = \frac{\text{RoPE}(\bar{\mathbf{C}}_k)}{\left\|\text{RoPE}(\bar{\mathbf{C}}_k)\right\|_2}, \quad \bar{\mathbf{B}}_k = \frac{\text{RoPE}(\bar{\mathbf{B}}_k)}{\left\|\text{RoPE}(\bar{\mathbf{B}}_k)\right\|_2} \tag{7}$$

Finally, the explicit affinity matrix is computed via the dot-product:

$$\text{Aff}_k = \bar{\mathbf{C}}_k \bar{\mathbf{B}}_k^\top, \quad \text{Aff}_k \in \mathbb{R}^{B \times N \times N} \tag{8}$$

This results in a structured affinity representation that preserves spatial relationships while maintaining computational efficiency. However, the computed affinity matrix $\text{Aff}_k$ (for each $k \in \{1, 2\}$) often exhibits dominant diagonal values due to high self-correlations. To mitigate this, we introduce a spatial suppression kernel:

$$K(i, j, i', j') = \frac{(i - i')^2 + (j - j')^2}{(i - i')^2 + (j - j')^2 + \sigma^2} \tag{9}$$

We apply it via element-wise modulation: $\text{Aff}'_k = \text{Aff}_k \odot K$, where $\odot$ denotes Hadamard product. To enforce mutual similarity, we adopt a bidirectional softmax strategy inspired by Cheng et al. (2019).

$$\text{Aff}_k^{\text{row}}(i, j) = \frac{\exp(\alpha \text{Aff}'_k[i, j])}{\sum_{j'} \exp(\alpha \text{Aff}'_k[i, j'])}, \quad \text{Aff}_k^{\text{col}}(i, j) = \frac{\exp(\alpha \text{Aff}'_k[i, j])}{\sum_{i'} \exp(\alpha \text{Aff}'_k[i', j])} \tag{10}$$

$$\text{Aff}_k^{\text{final}}(i, j) = \text{Aff}_k^{\text{row}}(i, j) \cdot \text{Aff}_k^{\text{col}}(i, j) \tag{11}$$

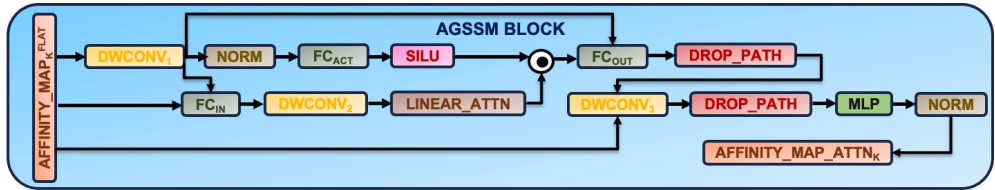

Figure 6: Affinity_Map$_k^{\text{flat}}$ is processed by the AGSSM Block to generate Affinity_Map_Attn$_k$

Here, $\alpha = 5$ is a temperature hyperparameter. The refined affinity matrix $\text{Aff}_k^{\text{final}} \in \mathbb{R}^{N \times N}$ is passed through a convolutional refinement module comprising four convolutional layers with SiLU activations, yielding the final affinity map Affinity_Map$_k \in \mathbb{R}^{\sqrt{N} \times \sqrt{N}}$, which highlights the top-$k$ most similar regions per spatial location.

**Affinity-Guided Self-Attention.** Each input feature map undergoes self-attention via three parallel Affinity-Guided modules operating on SSM-encoded features (AGSSM) in Figure 6. Each AGSSM block is modulated by a flattened affinity map Affinity_Map$_k^{\text{flat}} \in \mathbb{R}^N$, obtained by reshaping the Affinity_Map$_k$. This flattened representation is directly integrated into the self-attention mechanism as described below:

$$\text{Affinity\_Map}_k^{\text{flat}'} = \text{Affinity\_Map}_k^{\text{flat}} + \text{DWConv}_1(\text{Affinity\_Map}_k^{\text{flat}})$$
$$a = \text{SiLU}\big(\text{FC}_{\text{act}}(\text{Norm}(\text{Affinity\_Map}_k^{\text{flat}'}))\big)$$
$$x = \text{LinearAttn}\big(\text{DWConv}_2(\text{FC}_{\text{in}}(\text{Affinity\_Map}_k^{\text{flat}'})), \text{Affinity\_Map}_k^{\text{flat}}\big)$$
$$x = \text{Affinity\_Map}_k^{\text{flat}'} + \text{DropPath}\big(\text{FC}_{\text{out}}(x \odot a)\big)$$
$$\text{Affinity\_Map\_Attn}_k = x + \text{DWConv}_3(x) + \text{DropPath}\big(\text{MLP}(\text{Norm}(x))\big) \tag{12}$$

To promote diverse local interactions, three parallel AGSSM blocks are applied and averaged:

$$\text{AGSSM\_Self\_Attn}_k = \frac{1}{3}\sum_{i=1}^{3} \text{AGSSM}_i(\text{Affinity\_Map\_Attn}_k) \tag{13}$$

The aggregated representation is reshaped and projected using a $1 \times 1$ convolution:

$$\text{AGSSM\_Self\_Attn}_k^{\text{out}} = \text{Conv}_{1\times1}\big(\text{AGSSM\_Self\_Attn}_k^{\top}\big) \tag{14}$$

Affinity maps also enhance cross-view contextual consistency between paired inputs:

$$\text{AGSSM\_Self\_Attn}_1 = V_1 + \text{AGSSM\_Self\_Attn}_1^{\text{out}}$$
$$\text{AGSSM\_Self\_Attn}_2 = V_2 + \text{AGSSM\_Self\_Attn}_2^{\text{out}} \tag{15}$$

**Affinity-Guided Cross-Attention.** After self-attention enhancement, cross-attention is applied to promote interaction between paired features, guided by the projected affinity maps.

In the `CrossAttn` block, the inputs are first flattened, where $Q = \text{Flatten}(\text{AGSSM\_Self\_Attn}_1)$ and $K = \text{Flatten}(\text{Conv}_{1\times1}(\text{AGSSM\_Self\_Attn}_2)) \in \mathbb{R}^{B \times N \times C}$. Multi-head attention is computed as $\text{CrossAttn\_Out}_{\text{flat}} = \text{MultiHeadAttn}(Q, K, K)$, and the result is reshaped back to $\mathbb{R}^{B \times C \times H \times W}$.

$$V_1' = \text{AGSSM\_Self\_Attn}_1 + \text{CrossAttn}\big(\text{AGSSM\_Self\_Attn}_1, \text{Conv}_{1\times1}(\text{AGSSM\_Self\_Attn}_2)\big)$$
$$V_2' = \text{AGSSM\_Self\_Attn}_2 + \text{CrossAttn}\big(\text{AGSSM\_Self\_Attn}_2, \text{Conv}_{1\times1}(\text{AGSSM\_Self\_Attn}_1)\big) \tag{16}$$

**Proposition 1 (Cross-View Duplication Alignment Under Affinity-Guided Attention).** Let $V_1, V_2 \in \mathbb{R}^{N \times d}$ denote patch embeddings extracted from two image views. The affinity-guided cross-attention update is defined as:

$$V_1' = V_1 + \alpha\,(W_V V_2),$$

where $W_V \in \mathbb{R}^{d \times d}$ is the learned value-projection matrix and $\alpha = \text{Softmax}(A)$ with

$$A = (W_Q V_1)(W_K V_2)^\top + \Lambda,$$

where $W_Q, W_K \in \mathbb{R}^{d \times d}$ are the query/key projection matrices and $\Lambda$ is the affinity-guidance term derived from the AGSSM similarity module.

For a patch $i$ in $V_1$, suppose there exists a duplicated counterpart $j$ in $V_2$ such that the affinity-guided similarity satisfies the margin condition

$$A_{ij} \geq A_{ik} + \delta \qquad \forall k \neq j,$$

for some $\delta > 0$. Then the cross-attention update satisfies

$$\left\| V_1'(i) - \big(V_1(i) + W_V V_2(j)\big) \right\| \leq \epsilon,$$

where

$$\epsilon = \sum_{k \neq j} \alpha_{ik} \left\| W_V V_2(k) - W_V V_2(j) \right\|, \qquad \alpha = \text{Softmax}(A).$$

Thus, when the margin $\delta$ is sufficiently large, the affinity-guided cross-attention for patch $i$ is dominated by its duplicated counterpart $j$, yielding an update that is close to $V_1(i) + W_V V_2(j)$.

*Proof.* The cross-attention update for patch $i$ is

$$V_1'(i) = V_1(i) + \sum_{k=1}^{N} \alpha_{ik} W_V V_2(k), \qquad \alpha = \text{Softmax}(A).$$

Let $j$ be the duplicated patch corresponding to $i$. Subtracting and adding $W_V V_2(j)$ yields:

$$V_1'(i) - \big(V_1(i) + W_V V_2(j)\big) = \sum_{k \neq j} \alpha_{ik} \big(W_V V_2(k) - W_V V_2(j)\big).$$

Taking norms on both sides and using the triangle inequality:

$$\left\| V_1'(i) - (V_1(i) + W_V V_2(j)) \right\| \leq \sum_{k \neq j} \alpha_{ik} \left\| W_V V_2(k) - W_V V_2(j) \right\|.$$

It remains to analyze $\alpha_{ik}$. The affinity-guided score margin $A_{ij} \geq A_{ik} + \delta$ implies:

$$\alpha_{ij} = \frac{e^{A_{ij}}}{\sum_m e^{A_{im}}} \geq \frac{e^{A_{ik} + \delta}}{\sum_m e^{A_{im}}} = e^\delta \, \alpha_{ik}.$$

Thus, as $\delta$ increases, $\alpha_{ij}$ sharply dominates all $\alpha_{ik}$ for $k \neq j$, making the residual term $\epsilon$ small. Hence the update is numerically dominated by $W_V V_2(j)$, resulting in the stated bound. $\square$

### 3.4 SIAMESE DECODER

Each decoder branch maps the enriched features $V_1'$ and $V_2'$ to binary tampering masks using a lightweight convolutional head. Specifically, each feature $V_i'$ is processed through an intermediate stacked convolutional layers module $\phi$. This is followed by a $1 \times 1$ convolution and a sigmoid activation $\sigma$ to produce a single-channel probability map. The result is then upsampled via bilinear interpolation to match the original image resolution:

$$O_i = \text{Upsample}(\sigma(\text{Conv}_{1 \times 1}(\phi(V_i')))) \tag{17}$$

### 3.5 TRAINING DETAILS

Input images are resized to $224 \times 224$. Feature hierarchies span from $4096 \times 4096$ to $40 \times 40$.

Training minimizes a weighted sum of binary cross-entropy losses:

$$\mathcal{L} = w_{\text{self\_attn}} \mathcal{L}_{\text{BCE}}^{\text{self}} + w_{\text{cross\_attn}} \mathcal{L}_{\text{BCE}}^{\text{cross}} + w_{\text{fused}} \mathcal{L}_{\text{BCE}}^{\text{fused}} \tag{18}$$

where $\mathcal{L}_{\text{BCE}}$ denotes binary segmentation loss at each output stage.

We initialize with a ViT-Base encoder pretrained on ImageNet-1k and optimize using AdamW ($1 \times 10^{-4}$ learning rate). BioTamperNet is pretrained for 74 epochs on synthetic triplet patches and fine-tuned for 100 epochs on BioFors for External Duplication Detection (EDD). For Internal Duplication Detection (IDD) and Cut/Sharp Transition Detection (CSTD), images are split based on duplication masks. Early stopping and cosine learning rate decay are applied.

Table 1: Image and Pixel level MCC scores for EDD and IDD on the BioFors test set from retracted publications. All models are trained on the BioFors train set. **Best** in bold, Second best underlined.

| Method | External Duplication Detection (EDD) | | | | | | | | | | Internal Duplication Detection (IDD) | | | | | | | |
|---|---|---|---|---|---|---|---|---|---|---|---|---|---|---|---|---|---|---|
| | Microscopy | | Blot/Gel | | Macroscopy | | FACS | | Combined | | Microscopy | | Blot/Gel | | Macroscopy | | Combined | |
| | Image | Pixel | Image | Pixel | Image | Pixel | Image | Pixel | Image | Pixel | Image | Pixel | Image | Pixel | Image | Pixel | Image | Pixel |
| SIFT (IJCV 2004) | 0.180 | 0.146 | 0.113 | 0.148 | 0.130 | 0.194 | 0.110 | 0.073 | 0.142 | 0.132 | – | – | – | – | – | – | – | – |
| ORB (ICCV 2011) | 0.319 | 0.342 | 0.087 | 0.127 | 0.126 | 0.226 | 0.269 | 0.187 | 0.207 | 0.252 | – | – | – | – | – | – | – | – |
| BRIEF (ECCV 2010) | 0.275 | 0.277 | 0.058 | 0.102 | 0.135 | 0.169 | 0.244 | 0.188 | 0.180 | 0.202 | – | – | – | – | – | – | – | – |
| DF–ZM (TIFS 2015) | 0.422 | 0.425 | 0.161 | 0.192 | 0.285 | 0.256 | 0.540 | **0.504** | 0.278 | 0.324 | 0.764 | 0.197 | 0.515 | 0.449 | 0.573 | 0.478 | 0.564 | 0.353 |
| DMVN (ACM MM 2017) | 0.342 | 0.242 | 0.430 | 0.261 | 0.238 | 0.185 | 0.282 | 0.164 | 0.310 | 0.244 | – | – | – | – | – | – | – | – |
| DF–PCT (TIFS 2015) | – | – | – | – | – | – | – | – | – | – | 0.764 | 0.202 | 0.503 | 0.466 | 0.712 | 0.487 | 0.569 | 0.364 |
| DF–FMT (TIFS 2015) | – | – | – | – | – | – | – | – | – | – | 0.638 | 0.167 | 0.480 | 0.400 | 0.495 | 0.458 | 0.509 | 0.316 |
| BusterNet (ECCV 2018) | – | – | – | – | – | – | – | – | – | – | 0.183 | 0.178 | 0.226 | 0.076 | 0.021 | 0.106 | 0.269 | 0.107 |
| ManTraNet (CVPR 2019) | 0.347 | 0.244 | 0.449 | 0.287 | 0.275 | 0.202 | 0.337 | 0.186 | 0.351 | 0.231 | 0.316 | 0.194 | 0.317 | 0.094 | 0.272 | 0.262 | 0.335 | 0.183 |
| TruFor (CVPR 2023) | 0.381 | 0.264 | 0.485 | 0.311 | 0.257 | 0.358 | 0.363 | 0.194 | 0.371 | 0.282 | 0.336 | 0.199 | 0.351 | 0.106 | 0.324 | 0.297 | 0.337 | 0.201 |
| URN (Patterns 2024) | 0.322 | 0.235 | 0.347 | 0.252 | 0.273 | 0.180 | 0.270 | 0.160 | 0.303 | 0.207 | – | – | – | – | – | – | – | – |
| MONet (ICIP 2022) | 0.435 | 0.398 | 0.520 | 0.507 | 0.262 | 0.221 | 0.356 | 0.313 | 0.438 | 0.410 | – | – | – | – | – | – | – | – |
| SparseViT (AAAI 2025) | 0.384 | 0.269 | 0.482 | 0.309 | 0.271 | 0.371 | 0.376 | 0.202 | 0.378 | 0.288 | 0.342 | 0.211 | 0.356 | 0.117 | 0.331 | 0.305 | 0.343 | 0.211 |
| **BioTamperNet** | **0.739** | **0.487** | **0.672** | **0.589** | **0.743** | **0.577** | **0.652** | 0.448 | **0.701** | **0.526** | **0.827** | **0.526** | **0.681** | **0.617** | **0.843** | **0.679** | **0.701** | **0.534** |

## 4 EXPERIMENTAL EVALUATION

### 4.1 DATASET AND METRICS

We conduct our experiments on the BioFors Sabir et al. (2021) dataset, which includes 30,536 training images and 17,269 test images. The dataset is categorized into four distinct image types: Microscopy, Blot/Gel, Macroscopy, and Fluorescence-Activated Cell Sorting (FACS). BioTamperNet achieves state-of-the-art performance across all four categories as well as the combined set for the EDD, IDD and CSTD tasks, as shown in Table 1 and Appendix. Since the FACS subset does not contain internal duplication (IDD) cases, we evaluate BioTamperNet on the remaining three categories for IDD, as reported in Table 1. Each category presents unique semantic content and domain-specific visual characteristics, introducing diverse challenges for detection. We follow the evaluation protocol of Sabir et al. (2021), assessing performance at both the image and pixel levels. Image-level metrics are computed using binary labels per image, while pixel-level evaluation aggregates detection results across all pixels. We further evaluated two synthetic scientific integrity benchmarks - RSIID Cardenuto & Rocha (2022) (4,912 Microscopy test images), Western Blots Manjunath et al. (2024) (6,146 test images) and reported state-of-the-art performance in Table 2(c). We report the Matthews Correlation Coefficient (MCC) Chicco & Jurman (2020) for consistency with prior work.

### 4.2 RESULTS AND ANALYSIS

We achieve state-of-the-art performance on the BioFors Sabir et al. (2021) test set—comprising real-world duplication forgeries from retracted scientific publications—demonstrating robust results across EDD, IDD and CSTD tasks (Tables 1, 2(a)). Unlike MONet Sabir et al. (2022), which is limited to EDD and performs comparably to earlier baselines, BioTamperNet consistently outperforms across all biomedical image types—microscopy, blot/gel, FACS, macroscopy and all combined. Dense feature matching methods (DF-ZM, DF-PCT Cozzolino et al. (2015)) surpass traditional keypoint-descriptor techniques (DCT Fridrich et al. (2003), DWT Bashar et al. (2010), Zernike Ryu et al. (2010)) and low-level feature models like Trufor Guillaro et al. (2023), SparseViT Su et al. (2025), ManTra-Net Wu et al. (2019), though their efficacy is limited by repetitive biomedical patterns.

BioTamperNet is uniquely capable of localizing both duplicated regions and their corresponding sources, as illustrated in Figure 7—a capability rarely supported by existing forensic approaches. Conventional deep learning models Wu et al. (2019); Islam et al. (2020); Sabir et al. (2022); Guillaro et al. (2023); Su et al. (2025), primarily trained for splicing detection, are tailored to capture low-level manipulation artifacts. Consequently, they struggle with duplication-based manipulations in biomedical images. As shown in Figure 7(a), BioTamperNet accurately predicts both source and target regions in alignment with the ground-truth masks. In contrast, Wu et al. (2019) fails to distinguish duplication structure, capturing only low-level artifacts. Similarly, Sabir et al. (2021) and Guillaro et al. (2023) rely on patch-level comparisons and noiseprint features, causing frequent misdetections.

Figure 7(b) shows IDD results using both single-image and paired-image inputs. This dual evaluation is necessary because existing methods Wu et al. (2019); Islam et al. (2020); Guillaro et al. (2023) are designed to localize only target forgeries from single images, whereas BioTamperNet is explicitly built to localize both source and target regions using paired inputs. As observed, Guillaro et al. (2023) generates numerous false positives due to the misclassification of structured biomedical textures as tampered regions from noiseprint-based extracted features. Wu et al. (2019) captures superficial

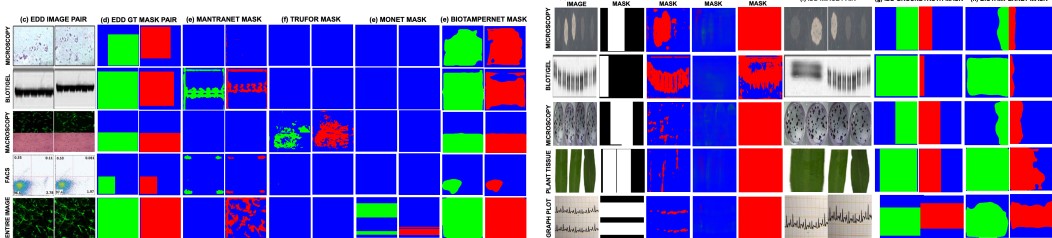

Figure 7: Qualitative results (a) EDD (b) IDD. Zoom in to view untampered, source, target regions.

Table 2: Additional Evaluation on CSTD, Classification and Scientific Integrity Synthetic Benchmarks (RSIID Cardenuto & Rocha (2022), Western Blots Manjunath et al. (2024))

**(a) CSTD Detection on BioFors**

| Method | F1 | | MCC | |
|---|---|---|---|---|
| | Image | Pixel | Image | Pixel |
| ManTraNet | 0.253 | 0.090 | 0.170 | 0.080 |
| TruFor | 0.311 | 0.104 | 0.173 | 0.092 |
| BioTamperNet | **0.537** | **0.378** | **0.514** | **0.346** |

**(b) Classification Accuracy**

| Model | Train | Test |
|---|---|---|
| ResNet | 98.93% | 97.47% |
| ViT-MedSam | 98.69% | 98.42% |
| ViT-ImageNet | **99.63%** | **99.54%** |

**(c) Scientific Integrity Synthetic (Pixel-level MCC)**

| Method | RSIID | Western Blots | Avg. |
|---|---|---|---|
| ManTraNet | 0.462 | 0.196 | 0.284 |
| TruFor | 0.825 | 0.457 | 0.553 |
| SparseViT | 0.842 | 0.739 | 0.612 |
| BioTamperNet | **0.965** | **0.913** | **0.628** |

low-level features without contextual understanding, and Islam et al. (2020) fails to detect coherent duplication patterns. In contrast, BioTamperNet robustly localizes both source and target regions with high spatial accuracy, even in challenging IDD scenarios.

This superior performance arises from BioTamperNet jointly modeling affinity-guided self-attention and duplication-aware cues—capabilities that are lacking in prior deep learning and traditional keypoint-descriptor-based methods, which primarily focus on RGB, frequency artifacts or boundary inconsistencies. For BioFors classification (Table 2(b)), the ImageNet-pretrained ViT achieves the highest accuracy, likely due to superior global context modeling leveraged in Eqn 4. BioTamperNet is the first to set baselines for IDD and CSTD and achieves state-of-the-art forgery detection with single and paired inputs across all three tasks and synthetic biomedical datasets (Table 2(c)).

## 4.3 ROBUSTNESS ANALYSIS

To assess resilience under perturbations, we evaluate BioTamperNet and competitors on BioFors Sabir et al. (2021). As shown in Figure 9, BioTamperNet consistently outperforms others, showing robustness to brightness change, JPEG compression, contrast adjustment, and noise. While BioTamperNet emphasizes localized contextual modeling, we also test a lightweight Global SSM layer after affinity-guided enrichment to extend long-range reasoning. Yet, since biomedical duplications are usually local or moderately displaced, global modeling adds little, yielding marginal changes (Table 3); further analysis is in the appendix.

## 4.4 ABLATION STUDIES

### 4.4.1 WHY BIOTAMPERNET IS TAILORED FOR BIOMEDICAL FORENSICS

BioTamperNet addresses unique challenges in biomedical image forensics, including structurally repetitive duplications (e.g., cell clusters, gel bands), low-texture environments, and modality inconsistencies that cause traditional copy-move detectors and ViT to overfit or forget previously learned patterns—akin to catastrophic forgetting. To mitigate this, BioTamperNet combines affinity-guided Siamese attention modules that explicitly model source-target duplication relationships with SSMs that offer superior long-range context modeling and convergence in low-data regimes. Unlike ViT or CNN backbones, SSMs maintain temporal continuity across spatial axes without requiring large training corpora. Additionally, domain-specific normalization stabilizes learning across microscopy, blot/gel, and macroscopy without separate models or modality-wise fine-tuning. We ablate key components of BioTamperNet to assess their role. Replacing the SSM with a 4-layer CNN (w/o SSM (CNN)) restricts the receptive field and weakens global consistency. Alternatively, substituting with a 4-layer ViT encoder (w/o SSM (ViT-MHA)) using multi-head attention and two-layer MLP blocks impairs convergence in data-sparse scenarios. As shown in Table 3, removing affinity or SSM

Table 3: MCC for BioTamperNet EDD Ablation.    Figure 8: Limitations in EDD, IDD, and CSTD.

| Model | Microscopy | Blot/Gel | Macroscopy |
|---|---|---|---|
| w/o Affinity | 0.421 | 0.489 | 0.462 |
| w/o SSM (CNN) | 0.393 | 0.453 | 0.437 |
| w/o SSM (ViT-MHA) | 0.407 | 0.466 | 0.445 |
| w/o Self-Attn | 0.451 | 0.509 | 0.492 |
| w/o Cross-Attn | 0.444 | 0.497 | 0.481 |
| w/ LayerNorm | 0.439 | 0.492 | 0.476 |
| **BioTamperNet (Full)** | **0.487** | **0.589** | 0.577 |
| **+ Global SSM** | 0.467 | 0.539 | **0.580** |

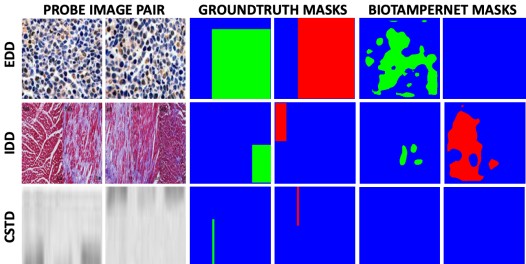

Table 4: MCC for Cross-Modality Fine-Tuning Ablation on EDD.

Figure 9: Forgery Robustness under attacks on BioFors.

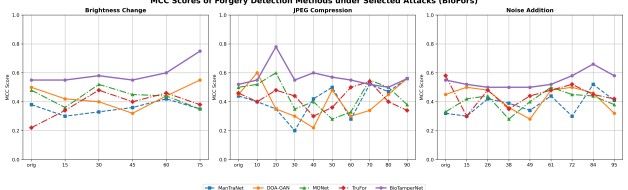

| Model | Source Microscopy | Target Blot/Gel | Avg. Retention |
|---|---|---|---|
| Naive Fine-Tuning | 0.387 | 0.409 | 0.398 |
| + Frozen Affinity | 0.430 | 0.424 | 0.427 |
| **BioTamperNet** | **0.487** | **0.589** | **0.538** |

components consistently reduces MCC across modalities, validating the architecture's effectiveness in preserving cross-modality generalization while avoiding collapse from domain shifts.

### 4.4.2 CROSS-MODALITY FINE-TUNING VS CATASTROPHIC FORGETTING

We investigate the effect of training BioTamperNet on one biomedical modality (e.g., microscopy) and fine-tuning on another (e.g., blot/gel), simulating domain expansion scenarios common in biomedical deployments. We compare three variants: (i) naive fine-tuning without batch normalization adaptation, (ii) fine-tuning with frozen affinity-guided modules but re-learned SSMs, and (iii) our full BioTamperNet with domain-adaptive batch normalization. Metrics are computed on both source and target domains to assess retention and adaptation. As shown in Table 4, naive fine-tuning suffers from catastrophic forgetting (–17.2% drop in source MCC), whereas our full model maintains balanced performance, confirming its biomedical robustness. Additional ablation results are in supplementary.

### 4.5 MODEL COMPLEXITY COMPARISON

BioTamperNet performs affinity computation on a compact $40 \times 40$ token grid ($N{=}1600$), where the full $N \times N$ operation adds only $\approx 1.0$ GFLOPs ($< 4\%$ of total cost). As shown in Table 5, the model uses 36.7M parameters and 29.6 GFLOPs for a $512{\times}512$ input, making it lighter than detectors such as TruFor and SparseViT while achieving stronger accuracy. This efficient design highlights BioTamperNet's scalability and suitability for high-fidelity and real-time forensic workloads.

### 4.6 LIMITATIONS AND FUTURE WORK

Despite strong performance, BioTamperNet exhibits three notable failure modes: (i) perfectly overlapping source–target regions where duplication boundaries vanish (Fig. 8, row 1); (ii) false positives in highly repetitive textures or clustered biomedical structures (row 2); and (iii) Blot/Gel images with dense stains and small-scale CSTD masks that reduce boundary contrast (row 3). To address these issues, we propose overlap-aware post-processing and auxiliary boundary heads for (i), enhanced with cycle-consistency and uniqueness regularizers and frequency-aware filtering to preserve duplication boundaries; semantic regularization via adaptive thresholding, entropy-based hard-negative mining, 2D RoPE integration, and localized non-max suppression for (ii) to suppress spurious matches in repetitive patterns; and stain-invariant augmentations combined with multi-scale feature refinement and grayscale-focused preprocessing for (iii) to improve detection of weak boundaries under chemical noise. Future work will extend BioTamperNet to video-level forgery detection by incorporating temporal attention and spatio-temporal consistency constraints.

Figure 10: BioTamperNet Affinity Heatmaps.

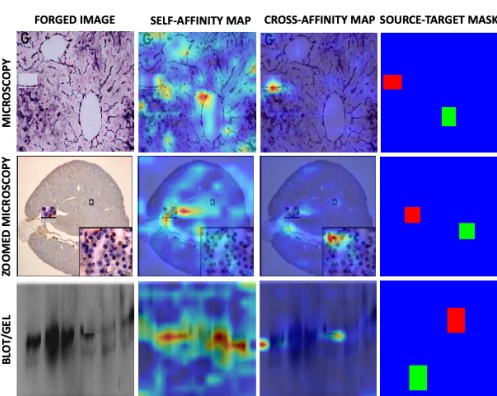

Table 5: Model complexity comparison.

| Method | Backbone | Input | Params | FLOPs |
|---|---|---|---|---|
| BusterNet | VGG | $256 \times 256$ | 15.5M | 45.7G |
| ManTraNet | VGG | $256 \times 256$ | 3.9M | 274.0G |
| MONet | ResNet | $512 \times 512$ | 31.0M | 78.0G |
| TruFor | ViT | $512 \times 512$ | 68.7M | 236.5G |
| SparseViT | ViT | $512 \times 512$ | 50.3M | 46.2G |
| **BioTamperNet** | SSM | $512 \times 512$ | **36.7M** | **29.6G** |

The affinity heatmaps in Fig. 10 offer an interpretable view by showing: (1) self-affinity revealing similar structures, (2) cross-affinity highlighting source–target duplication links, and (3) ground-truth untampered, source, and target regions across biomedical modalities. These cues demonstrate BioTamperNet's ability to find similar patterns and refine duplicate detection.

## 5 RELATED WORK

**Biomedical Image Forensics and Research Misrepresentation.** Image manipulation in biomedical research is serious, with 3.8% papers containing at least one case Bik et al. (2016). Manual corrections exist, but the growing volume of publications makes this unsustainable. Computer vision has been widely applied to medical imaging tasks such as segmentation Shamshad et al. (2023); Heidari et al. (2023); Wu et al. (2022), super-resolution Jeelani et al. (2023), and disease diagnosis Esteva et al. (2021). In contrast, its use in detecting fraudulent scientific content—has received limited attention. This gap is largely due to the scarcity of large-scale annotated datasets, which hinders the development of robust deep learning models. Early efforts in biomedical forensics include duplication screening Koppers et al. (2017), copy-move forgery detection (CMFD) Bucci (2018), and SIFT-based matching Acuna et al. (2018), but lacking standardized benchmarks like TrainFors Nandi et al. (2023).

**Duplication Detection in Natural and Biomedical Images.** Traditional forgery detectors Lowe (2004); Rublee et al. (2011); Calonder et al. (2010); Fridrich et al. (2003) work well on natural images but struggle with the subtle textures of biomedical data. Deep splicing in Wu et al. (2019); Sabir et al. (2022); Guillaro et al. (2023) and copy-move models Wu et al. (2018); Islam et al. (2020) still rely mainly on low-level cues, while dense feature matching Cozzolino et al. (2015) remains computationally expensive. BioFors Sabir et al. (2021) introduced unified benchmarks for EDD, IDD, and CSTD, yet recent systems—e.g., MONet Sabir et al. (2022) focused on external duplications and URN Lin et al. (2024) tailored for splicing—offer limited transfer to the diverse, duplication-centric manipulations found in biomedical figures. In contrast, BioTamperNet provides a purpose-built, unified architecture that handles both source and duplicated regions across modalities with significantly improved accuracy and efficiency, addressing the shortcomings of prior approaches.

**State Space Models for Vision Tasks.** Structured State Space Models (SSMs) such as S4 Gu et al. (2021) and Mamba Gu & Dao (2023), Mamba2 Dao & Gu (2024) model long-range dependencies efficiently and have recently been extended to vision applications Zhu et al. (2024); Liu et al. (2024). MILA Han et al. (2024) further analyzes design factors—including gating and block structure—that distinguish effective SSMs from linear-attention Transformers. However, these general-purpose models lack the task-specific structure needed for fine-grained forgery analysis. We adapt SSMs for duplication localization, yielding improved accuracy and efficiency for biomedical forensics.

## 6 CONCLUSION

We presented BioTamperNet, a unified framework for detecting both external and internal duplications in biomedical images. By leveraging affinity-guided self-attention and cross-attention modules built on linear approximations of State Space Models (SSMs), BioTamperNet efficiently captures fine-grained correspondences and semantic inconsistencies across paired views. Extensive experiments on retracted and synthetic datasets demonstrate that our method achieves state-of-the-art localization performance while remaining lightweight and modular. Pretraining on adversarially augmented synthetic tampering datasets further improves performance on BioFors by enhancing robustness to diverse manipulation patterns. Our approach simplifies forgery detection by unifying paired-image and single-image localization within a single architecture.

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
