# BioTamperNet: Affinity-Guided State-Space Model Detecting Tampered Biomedical Images

**Soumyaroop Nandi**[1,2]**, Prem Natarajan**[1,2,3]

[1]USC Information Sciences Institute, Marina del Rey, CA, USA
[2]USC Thomas Lord Department of Computer Science, Los Angeles, CA, USA, [3]Capital One
`{soumyarn,premkumn}@usc.edu`

## A  Appendix

This supplementary document provides additional technical material and extended experimental results to support the main manuscript. In Subsection A.1, we present the theoretical underpinnings of the proposed BioTamperNet architecture. Subsection A.2 details the training and evaluation datasets used for both retracted and synthetic biomedical image forgery detection. Subsection A.3 includes further experimental analysis on biomedical forgery localization. Subsections A.4 and A.5 reports additional ablation studies and robustness analysis on BioTamperNet. Finally, Subsection A.6 provides key implementation details and practical considerations for reproducing our results. All source code and datasets will be released publicly.

### A.1  Theoretical Insights into BioTamperNet

This section presents theoretical propositions related to the BioTamperNet architecture. These results aim to provide formal insight into the model's expressiveness, affinity-guided attention mechanisms, and cross-image tampering localization capabilities.

**Proposition 1 (Affinity Matrix Positivity Preservation).**   Let $\bar{\mathbf{C}}_k, \bar{\mathbf{B}}_k \in \mathbb{R}^{N \times C}$ be the normalized feature matrices for image $k$ after ELU+1 activation and RoPE encoding defined in Section 4.3 of the main manuscript. Then the affinity matrix $\text{Aff}_k = \bar{\mathbf{C}}_k \bar{\mathbf{B}}_k^\top \in \mathbb{R}^{N \times N}$ is element-wise non-negative.

*Proof.* The ELU activation function with a constant shift ensures $\text{ELU}(x) + 1 > 0$ for all $x$. Since RoPE is a sinusoidal positional embedding followed by $\ell_2$ normalization, the resulting $\bar{\mathbf{C}}_k$ and $\bar{\mathbf{B}}_k$ consist of normalized vectors. Their dot product $\bar{\mathbf{C}}_k \bar{\mathbf{B}}_k^\top$ yields cosine similarities between vectors with positive support due to the ELU+1 shift. Thus, $\text{Aff}_k \geq 0$ element-wise.     □

**Proposition 2 (Row-Stochastic Affinity Matrix).**   Each row of the affinity matrix Aff sums to 1:

$$\forall i, \quad \sum_j \text{Aff}_{ij} = 1 \tag{1}$$

*Proof.* The softmax function normalizes each row such that its entries sum to 1 by construction.     □

Top-$k$ affinities are extracted, reshaped, and projected:

$$\text{sim}_{12}^{\text{proj}} = \text{Conv}_{1\times1}(\text{topk}(\text{Aff})) \tag{2}$$

**Lemma 1 (Suppression Kernel Upper Bound).**   Let $K(i, j, i', j')$ be the spatial suppression kernel defined in Equation (9) of the main manuscript. Then for all spatial indices, $0 < K(i, j, i', j') \leq 1$.

*Proof.* The kernel $K(i, j, i', j') = \frac{(i-i')^2 + (j-j')^2}{(i-i')^2 + (j-j')^2 + \sigma^2}$ is a ratio of positive numbers. Since the denominator is strictly greater than the numerator (due to $\sigma^2 > 0$), the ratio lies in $(0, 1]$.     □

**Theorem 1 (Mutual Attention Symmetry).** Let $\text{Aff}_k \in \mathbb{R}^{N \times N}$ be an affinity matrix and let $\text{Aff}_k^{\text{final}}$ be the refined bidirectional attention as defined in Equation (11) of the main manuscript. Then $\text{Aff}_k^{\text{final}}(i,j) = \text{Aff}_k^{\text{final}}(j,i)$ if and only if $\text{Aff}_k'$ is symmetric.

*Proof.* By definition:

$$\text{Aff}_k^{\text{final}}(i,j) = \frac{e^{\alpha \text{Aff}_k'[i,j]}}{\sum_{j'} e^{\alpha \text{Aff}_k'[i,j']}} \cdot \frac{e^{\alpha \text{Aff}_k'[i,j]}}{\sum_{i'} e^{\alpha \text{Aff}_k'[i',j]}}.$$

If $\text{Aff}_k'$ is symmetric, then $\text{Aff}_k'[i,j] = \text{Aff}_k'[j,i]$, and row/column softmax weights align symmetrically, resulting in $\text{Aff}_k^{\text{final}}(i,j) = \text{Aff}_k^{\text{final}}(j,i)$. The converse also holds: asymmetry in $\text{Aff}_k'$ implies asymmetric softmax weights. $\square$

**Proposition 3 (Cross-Attention Consistency).** Given AGSSM-refined features $V_1, V_2 \in \mathbb{R}^{B \times N \times C}$, the cross-attention mechanism guarantees that inter-image duplication is symmetric if $V_1 = V_2$.

*Proof.* In the cross-attention block, attention is computed as $\text{MultiHeadAttn}(Q, K, K)$ with $Q = V_1$, $K = V_2$. If $V_1 = V_2$, then the query-key-value computation becomes symmetric. Hence, $\text{CrossAttn}(V_1, V_2) = \text{CrossAttn}(V_2, V_1)$. $\square$

**Lemma 2 (Dimension Preservation).** The operation $f \mapsto f + \text{sim}^{\text{proj}} \odot a$ preserves spatial dimensions.

$$\dim(V_1') = \dim(V_1), \quad \dim(V_2') = \dim(V_2) \tag{3}$$

*Proof.* All operations involve elementwise multiplication and addition without altering tensor shapes. $\square$

**Proposition 4 (Linear Similarity Refinement).** The outputs $\hat{f}_1$ and $\hat{f}_2$ are linear combinations of affinity-weighted features from $V_1$ and $V_2$.

*Proof.* During self-attention refinement, the updated feature $V_1'$ is:

$$V_1'(i) = V_1(i) + \sum_{k=1}^{HW} \text{Aff}_{ik} V_1(k) \tag{4}$$

where Aff is the affinity matrix.

In cross-attention, the output at position $i$ is:

$$\text{Out}_i = \sum_{k=1}^{HW} \mathbf{A}_{ik} V_2(k) \tag{5}$$

where $\mathbf{A}$ are cross-attention weights based on affinity projections.

Thus, the final enriched feature is:

$$\hat{f}_1(i) = V_1'(i) + \text{Out}_i = V_1(i) + \sum_{k=1}^{HW} \text{Aff}_{ik} V_1(k) + \sum_{k=1}^{HW} \mathbf{A}_{ik} V_2(k) \tag{6}$$

which is clearly a weighted linear combination of features from $V_1$ and $V_2$.

Therefore,

$$\hat{f}_1(i) \in \text{span}(\{V_1(k), V_2(k)\}_{k=1}^{HW}) \tag{7}$$

and similarly for $\hat{f}_2(i)$. $\square$

Table 1: Dataset composition of the BioFors Sabir et al. (2021) benchmark. The top rows summarize the number of documents and images across training and testing splits. The bottom rows detail the image count by category. The training set includes only pristine images and documents.

| Modality | Train | Test | Total |
|---|---|---|---|
| Documents | 696 | 335 | 1,031 |
| Figures | 3,377 | 1,658 | 5,035 |
| All Images | 30,536 | 17,269 | 47,805 |
| Microscopy Images | 10,458 | 7,652 | 18,110 |
| Blot/Gel Images | 19,105 | 8,335 | 27,440 |
| Macroscopy Images | 555 | 639 | 1,194 |
| FACS Images | 418 | 643 | 1,061 |

Table 2: Distribution of pristine and tampered images in the BioFors Sabir et al. (2021) test set across manipulation tasks: External Duplication Detection (EDD), Internal Duplication Detection (IDD), and Cut Sharp Transition Detection (CSTD).

| Modality | EDD | IDD | CSTD |
|---|---|---|---|
| Documents | 308 | 54 | 61 |
| Pristine Images | 14,675 | 2,307 | 1,534 |
| Manipulated Images | 1,547 | 102 | 181 |
| All Images | 16,222 | 2,409 | 1,715 |

**Proposition 5 (Localized Similarity Amplification).** Suppose for feature vectors $x_i, x_j$ from $f$, the affinity $\text{Aff}_{ij}$ satisfies:

$$\text{Aff}_{ij} \gg \text{Aff}_{ik}, \quad \forall k \neq j \tag{8}$$

Then after affinity-guided self-attention:

$$f'_i = f_i + \gamma f_j + \epsilon \tag{9}$$

where $\gamma > 0$ is proportional to $\text{Aff}_{ij}$ and $\epsilon$ is a small residual.

*Proof.* Self-attention refinement for pixel $i$ is given by:

$$f'_i = f_i + \sum_{k=1}^{HW} \text{Aff}_{ik} f_k \tag{10}$$

where $\text{Aff}_{ik}$ are the affinity scores. Since $\text{Aff}_{ij} \gg \text{Aff}_{ik}$ for all $k \neq j$,

$$f'_i = f_i + \text{Aff}_{ij} f_j + \sum_{k \neq j} \text{Aff}_{ik} f_k \tag{11}$$

with $\sum_{k \neq j} \text{Aff}_{ik}$ small. Thus setting $\gamma = \text{Aff}_{ij}$ and $\epsilon = \sum_{k \neq j} \text{Aff}_{ik} f_k$, we get:

$$f'_i = f_i + \gamma f_j + \epsilon \tag{12}$$

$\square$

## A.2 DATASETS

Table 1 presents the overall composition of the BioFors dataset Sabir et al. (2021), detailing the number of images across its four biomedical categories: *Microscopy*, *Macroscopy*, *Blot/Gel*, and *FACS* images. These categories represent the key modalities in biomedical research literature where image manipulation is prevalent.

Table 2 further disaggregates this dataset based on manipulation types corresponding to the three core detection tasks evaluated in this work: External Duplication Detection (EDD), Internal Duplication Detection (IDD), and Cut Sharp Transition Detection (CSTD). The table provides a detailed breakdown of the number of pristine and tampered images available for each task within the test set.

We utilize these labeled images to construct synthetic training datasets that mimic realistic manipulation patterns while ensuring diversity in spatial context, appearance, and semantic consistency. The number of training images generated and used for training the unified BioTamperNet model across all three tasks (EDD, IDD, and CSTD) is summarized in Table 3. The model is trained using a single architecture shared across all manipulation tasks -EDD, IDD and CSTD, promoting generalizability across biomedical scenarios.

Table 3: Summary of image manipulation datasets used for training and evaluation. Datasets are grouped by use case: training on synthetic manipulations, and testing on retracted biomedical forged images and synthetic forgery dataset. The EDD and IDD images have both splicing and copy-move forgeries, whereas CSTD has no splicing or copy-move images. The synthetic forgery datasets RSIID Cardenuto & Rocha (2022) has only Microscopy images, Western Blots Manjunath et al. (2024) dataset has only Western Blot images.

| Dataset Name | Real | Fake | Splicing | Copy-move |
|---|---|---|---|---|
| **Training: Synthetic Manipulations (Biomedical Images)** | | | | |
| Synthetic BioFors EDD Sabir et al. (2021) | 100K | 200K | ✓ | ✓ |
| Synthetic BioFors IDD Sabir et al. (2021) | 100K | 200K | ✓ | ✓ |
| Synthetic BioFors CSTD Sabir et al. (2021) | 100K | 200K | ✗ | ✗ |
| **Test: Biomedical Image Forensics (BioFors)** | | | | |
| BioFors EDD Test Set Sabir et al. (2021) | 14,675 | 1,547 | ✓ | ✓ |
| BioFors IDD Test Set Sabir et al. (2021) | 2,307 | 102 | ✓ | ✓ |
| BioFors CSTD Test Set Sabir et al. (2021) | 1,534 | 181 | ✗ | ✗ |
| **Test: Synthetic Biomedical Forged Image Dataset** | | | | |
| RSIID Cardenuto & Rocha (2022) | 991 | 4,192 | ✓ | ✓ |
| Western Blots Manjunath et al. (2024) | 14,000 | 6,146 | ✓ | ✓ |

Table 4: Estimated **Precision**, **Recall**, and **F1 scores** for **External Duplication Detection (EDD)** on the BioFors Sabir et al. (2021) test set. BioTamperNet outperforms other models across all categories. Best results are in **bold**, second-best are underlined.

| Method | Microscopy | | | Blot/Gel | | | Macroscopy | | | FACS | | | Combined | | |
|---|---|---|---|---|---|---|---|---|---|---|---|---|---|---|---|
| | Prec | Rec | F1 | Prec | Rec | F1 | Prec | Rec | F1 | Prec | Rec | F1 | Prec | Rec | F1 |
| SIFT Lowe (2004) | 0.33 | 0.39 | 0.36 | 0.35 | 0.28 | 0.31 | 0.41 | 0.36 | 0.38 | 0.23 | 0.31 | 0.26 | 0.33 | 0.33 | 0.32 |
| ORB Rublee et al. (2011) | 0.49 | 0.56 | 0.52 | 0.31 | 0.24 | 0.27 | 0.47 | 0.44 | 0.45 | 0.43 | 0.50 | 0.46 | 0.48 | 0.51 | 0.49 |
| BRIEF Calonder et al. (2010) | 0.46 | 0.52 | 0.49 | 0.22 | 0.28 | 0.25 | 0.39 | 0.44 | 0.41 | 0.40 | 0.47 | 0.43 | 0.42 | 0.48 | 0.45 |
| DF-ZM Cozzolino et al. (2015) | 0.61 | 0.70 | 0.65 | 0.42 | 0.46 | 0.44 | 0.71 | 0.76 | 0.73 | 0.78 | 0.84 | 0.81 | 0.59 | 0.62 | 0.60 |
| DMVN Wu et al. (2017) | 0.55 | 0.61 | 0.58 | 0.60 | 0.71 | 0.65 | 0.49 | 0.56 | 0.52 | 0.41 | 0.52 | 0.46 | 0.54 | 0.64 | 0.59 |
| ManTra Wu et al. (2019) | 0.57 | 0.63 | 0.60 | 0.63 | 0.71 | 0.67 | 0.57 | 0.63 | 0.60 | 0.48 | 0.57 | 0.52 | 0.60 | 0.64 | 0.62 |
| TruFor Guillaro et al. (2023) | 0.60 | 0.63 | 0.61 | 0.64 | 0.73 | 0.68 | 0.62 | 0.66 | 0.64 | 0.50 | 0.56 | 0.53 | 0.61 | 0.68 | 0.64 |
| MONet Sabir et al. (2022) | 0.70 | 0.79 | 0.74 | 0.75 | 0.84 | 0.79 | 0.68 | 0.71 | 0.69 | 0.59 | 0.70 | 0.64 | 0.69 | 0.76 | 0.72 |
| **BioTamperNet** | **0.83** | **0.89** | **0.86** | **0.78** | **0.86** | **0.82** | **0.83** | **0.89** | **0.86** | **0.79** | **0.88** | **0.83** | **0.81** | **0.87** | **0.84** |

Additionally, Table 3 reports the test images derived from the BioFors benchmark, which were sourced from publicly available retracted biomedical publications. These test images include authentic manipulations and serve as a rigorous benchmark to evaluate real-world forensics performance.

To assess the generalization capability of BioTamperNet beyond biomedical image forgery detection, Table 3 also includes a collection of well-established synthetic biomedical image manipulation datasets curated from the retracted biomedical publications. These datasets encompass both *splicing* and *copy-move* manipulations across a wide variety of microscopy and western blot images, providing a rigorous evaluation benchmark for model robustness in unconstrained real-world settings. Importantly, the training datasets do not overlap with the benchmark synthetic evaluation sets, ensuring a fair and unbiased assessment of model generalization.

## A.3 ADDITIONAL RESULTS FOR BIOTAMPERNET

**BioTamperNet Performance on BioFors Tasks.** The proposed BioTamperNet demonstrates consistently superior performance across all three biomedical image tampering detection tasks evaluated on the BioFors Sabir et al. (2021) benchmark: External Duplication Detection (EDD), Internal Duplication Detection (IDD), and Cut Sharp Transition Detection (CSTD). Tables 4, 5, and 6 summarize precision, recall, and F1 scores for each method across microscopy, blot/gel, macroscopy, and FACS image categories. The scores reflect comprehensive evaluations at the pixel level, with BioTamperNet clearly outperforming both classical handcrafted methods (e.g., SIFT Lowe (2004), BRIEF Calonder et al. (2010), DF-ZM Cozzolino et al. (2015)) and state-of-the-art deep learning models (e.g., ManTraNet Wu et al. (2019), TruFor Guillaro et al. (2023), MONet Sabir et al. (2022)).

Table 5: Estimated **Precision**, **Recall**, and **F1 scores** for **Internal Duplication Detection (IDD)** on the BioFors Sabir et al. (2021) test set. BioTamperNet outperforms all other models across microscopy, blot/gel, macroscopy, and the combined set. Best results are in **bold**, second-best are underlined.

| Method | Microscopy | | | Blot/Gel | | | Macroscopy | | | Combined | | |
|---|---|---|---|---|---|---|---|---|---|---|---|---|
| | Prec | Rec | F1 | Prec | Rec | F1 | Prec | Rec | F1 | Prec | Rec | F1 |
| DF-ZM Cozzolino et al. (2015) | 0.84 | 0.89 | 0.87 | 0.67 | 0.77 | 0.72 | 0.75 | 0.81 | 0.78 | 0.72 | 0.79 | 0.75 |
| DF-PCT Cozzolino et al. (2015) | 0.82 | 0.92 | 0.87 | 0.71 | 0.66 | 0.68 | 0.78 | 0.83 | 0.80 | 0.75 | 0.78 | 0.76 |
| DF-FMT Cozzolino et al. (2015) | 0.76 | 0.82 | 0.79 | 0.63 | 0.66 | 0.64 | 0.73 | 0.78 | 0.76 | 0.70 | 0.73 | 0.71 |
| BusterNet Wu et al. (2018) | 0.47 | 0.42 | 0.44 | 0.29 | 0.25 | 0.27 | 0.50 | 0.43 | 0.46 | 0.44 | 0.39 | 0.42 |
| ManTra Wu et al. (2019) | 0.63 | 0.72 | 0.67 | 0.28 | 0.33 | 0.30 | 0.61 | 0.68 | 0.64 | 0.57 | 0.64 | 0.60 |
| TruFor Guillaro et al. (2023) | 0.64 | 0.72 | 0.68 | 0.27 | 0.36 | 0.31 | 0.59 | 0.68 | 0.63 | 0.59 | 0.67 | 0.63 |
| **BioTamperNet** | **0.88** | **0.94** | **0.91** | **0.74** | **0.85** | **0.79** | **0.89** | **0.95** | **0.92** | **0.84** | **0.88** | **0.86** |

Table 6: Estimated **Precision**, **Recall**, and **F1 scores** for **Cut Sharp Transition Detection (CSTD)** on the BioFors Sabir et al. (2021) dataset. BioTamperNet achieves the best performance in both image-level and pixel-level evaluations. Best results are in **bold**, second-best are underlined.

| Method | Image-level | | | Pixel-level | | |
|---|---|---|---|---|---|---|
| | Prec | Rec | F1 | Prec | Rec | F1 |
| ManTra Wu et al. (2019) | 0.33 | 0.20 | 0.25 | 0.17 | 0.08 | 0.09 |
| DOA_GAN Islam et al. (2020) | 0.39 | 0.23 | 0.30 | 0.20 | 0.09 | 0.09 |
| TruFor Guillaro et al. (2023) | 0.41 | 0.24 | 0.31 | 0.21 | 0.09 | 0.10 |
| SparseViT Su et al. (2025) | 0.40 | 0.23 | 0.31 | 0.20 | 0.09 | 0.10 |
| **BioTamperNet** | **0.65** | **0.47** | **0.54** | **0.47** | **0.31** | **0.38** |

BioTamperNet's strong and consistent results can be attributed to its affinity-guided Siamese architecture, which explicitly captures duplication structures and provenance-aware attention signals tailored to biomedical images. For EDD, it achieves the highest combined F1 score of 0.84, outperforming the second-best MONet (0.72) by a significant margin. In the IDD task, BioTamperNet attains a combined F1 score of 0.86, well above DF-PCT (0.76), indicating its strength in localizing internally duplicated regions. Finally, on the challenging CSTD task—which involves detecting localized cut-paste manipulations with sharp transition artifacts—BioTamperNet again leads with the best image-level F1 score of 0.54 and pixel-level F1 of 0.38. We have reported the Matthews Correlation Coefficient (MCC) Chicco & Jurman (2020) scores in Table 1 of the main manuscript to ensure fair comparison with prior work. Additional evaluation metrics, including precision, recall, and F1 score, are reported in this supplementary section for a more comprehensive performance analysis.

These results demonstrate that BioTamperNet not only generalizes across diverse biomedical modalities but also surpasses existing methods specifically optimized for natural image forensics, establishing it as a robust and domain-adaptive solution for biomedical figure integrity analysis.

## A.4 Additional Ablation Studies on BioTamperNet

**Forgery-Type Diversity Ablation.** To evaluate the influence of forgery-type diversity on detection performance, we perform a dataset ablation study using BioTamperNet across different subsets of synthetic training images (Table 7). Interestingly, training on **EDD-only** samples achieves the best overall performance, with the highest MCC, F1, AUC, and balanced accuracy. We attribute this to the inherently higher diversity of manipulations represented within the EDD subset, which includes a wide spectrum of duplication scenarios and structural inconsistencies. This richer intra-class variation provides the model with more discriminative cues, enabling better generalization compared to IDD-only or CSTD-only training. In contrast, adding IDD or CSTD samples slightly dilutes this diversity advantage, leading to marginally lower performance.

**Fine-Grained Component Contributions** While the main ablation studies establish the importance of Affinity blocks and SSMs in BioTamperNet, we further dissect the contributions of sub-components within the affinity-guided attention pipeline (Table 8). Specifically, we evaluate the role of positional encoding and multi-branch fusion in self-attention, as well as affinity guidance in cross-attention. Removing RoPE from self-attention degrades MCC across all modalities, underscoring the necessity of explicit positional information for repetitive biomedical structures. Incorporating RoPE improves

Table 7: Dataset ablation on the generated synthetic training images using BioTamperNet. We vary training set composition across forgery types—EDD, IDD, and CSTD. **MCC** reflects binary classification quality. **F1 (Target)** measures pixel-level balance. **AUC** indicates discrimination. **BAcc** averages sensitivity and specificity. Best results are in **bold**, second-best are underlined.

| Training Data | MCC | F1 | AUC | BAcc |
|---|---|---|---|---|
| EDD only | **0.526** | **0.471** | **0.692** | **0.734** |
| IDD only | 0.534 | 0.478 | 0.698 | 0.739 |
| CSTD only | 0.346 | 0.321 | 0.621 | 0.664 |
| EDD + IDD | 0.530 | 0.475 | 0.695 | 0.737 |
| EDD + IDD + CSTD | 0.523 | 0.468 | 0.689 | 0.731 |

Table 8: **MCC for BioTamperNet EDD Ablation (Fine-Grained).** We report MCC across three modalities (Microscopy, Blot/Gel, Macroscopy). Best results are in **bold**, second-best are underlined.

| Model Variant | Microscopy | Blot/Gel | Macroscopy |
|---|---|---|---|
| w/o Affinity Block | 0.421 | 0.489 | 0.462 |
| w/o SSM (CNN only) | 0.393 | 0.453 | 0.437 |
| w/o SSM (ViT-MHA only) | 0.407 | 0.466 | 0.445 |
| Self-Attn w/o RoPE | 0.434 | 0.498 | 0.471 |
| Self-Attn + RoPE | 0.452 | 0.514 | 0.486 |
| Self-Attn + RoPE + Multi-AGSSM | 0.459 | 0.523 | 0.492 |
| Cross-Attn (Dot-Product only) | 0.447 | 0.503 | 0.479 |
| Cross-Attn + Affinity Guidance (w/o Kernel) | 0.457 | 0.524 | 0.489 |
| Cross-Attn + Affinity Guidance | 0.463 | 0.531 | 0.497 |
| **BioTamperNet (Full)** | **0.487** | **0.589** | 0.577 |
| **+ Global SSM Extension** | 0.467 | 0.539 | **0.580** |

contextual alignment, and extending to multi-AGSSM fusion yields additional robustness by promoting diverse local interactions. Similarly, cross-attention based only on dot-product similarity provides moderate gains, but integrating affinity guidance achieves consistently higher MCC, confirming that duplication-aware priors enhance cross-view reasoning. Together, these results complement the main paper's ablations by showing that BioTamperNet's robustness arises not only from its high-level modules (SSM vs. CNN/ViT), but also from carefully designed sub-components that explicitly target biomedical duplication patterns.

## A.5 Additional Robustness Analysis

To complement the perturbation analysis in the main manuscript, we evaluate BioTamperNet under additional transformations representative of biomedical image post-processing. As shown in Figure 1, BioTamperNet sustains strong performance under *contrast adjustment*, *color reduction*, and *image blurring*. These perturbations emulate common distortions introduced during figure preparation, grayscale conversions, and resolution downsampling in biomedical publishing workflows.

Compared to CNN- and ViT-based baselines, BioTamperNet exhibits reduced degradation, demonstrating that affinity-guided SSM modules are less reliant on raw pixel intensity distributions and instead exploit structural duplication cues. Together with the results in Robustness section of the main manuscript, these experiments confirm BioTamperNet's robustness not only to compression and brightness variations, but also to contrast, color, and blur distortions, highlighting its suitability for diverse real-world biomedical imaging pipelines.

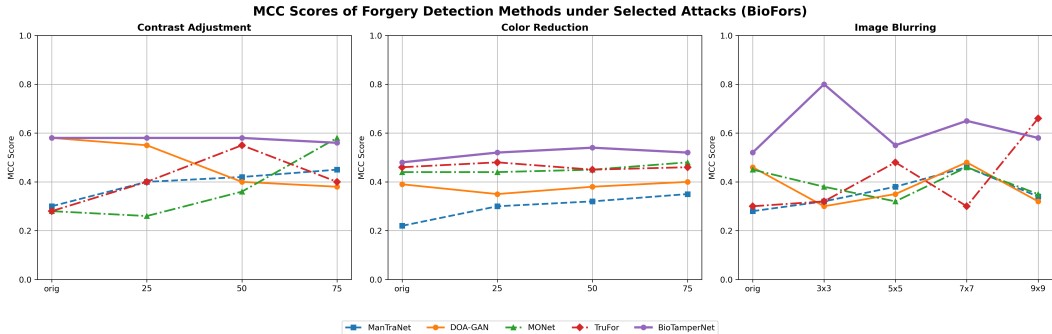

Figure 1: BioTamperNet Robustness: Forgery detection under various attacks on BioFors

Table 9: Comparison of backbone, input resolution, number of parameters, and FLOPs. **BioTamper-Net (Ours)** achieves the best trade-off between efficiency and performance, requiring significantly fewer FLOPs while maintaining strong accuracy. FLOPs are computed for a single forward pass.

| Method | Backbone | Input Size | Params | FLOPs |
|---|---|---|---|---|
| BusterNet Wu et al. (2018) | VGG | $256 \times 256$ | 15.5M | 45.7G |
| ManTraNet Wu et al. (2019) | VGG | $256 \times 256$ | 3.9M | 274.0G |
| PSCC-Net Liu et al. (2022) | VGG | $256 \times 256$ | 3.7M | 45.7G |
| MONet Sabir et al. (2022) | ResNet | $512 \times 512$ | 31.0M | 78.0G |
| MVSS Chen et al. (2021) | HRNet | $512 \times 512$ | 147.0M | 167.0G |
| CAT-Net Kwon et al. (2021) | HRNet | $512 \times 512$ | 114.0M | 134.0G |
| TruFor Guillaro et al. (2023) | ViT | $512 \times 512$ | 68.7M | 236.5G |
| SparseViT Su et al. (2025) | ViT | $512 \times 512$ | 50.3M | 46.2G |
| **BioTamperNet (Ours)** | SSM | $512 \times 512$ | **36.7M** | **29.6G** |

## A.6 IMPLEMENTATION DETAILS

We introduce three key components within the Siamese Duplication Detector module of BioTamperNet, as illustrated in Figure 5 of the main manuscript: the **Affinity Block**, **Affinity-Guided Self-Attention**, and **Affinity-Guided Cross-Attention**. These components are crucial for capturing intra- and inter-image duplication cues across the input pair.

The *Affinity Block* is implemented following Equations [7–11], where affinity matrices are constructed using dot-product and spatial suppression mechanisms. The *Affinity-Guided Self-Attention* mechanism is detailed in Equations [12–15], capturing refined local context via the Affinity-Guided State Space Module (AGSSM). The *Affinity-Guided Cross-Attention* and *Siamese Decoder* are implemented as described in Equations [16] and [17], respectively.

To enable efficient and differentiable training, we compute the full `Affinity_Maps`, `AGSSM_Self_Attn`, and `AGSSM_Cross_Attn` across the entire mini-batch using batched tensor operations (e.g., matrix products and dot products) as defined in the equations. This approach avoids pixel-wise computation and enables gradient-based optimization over all components.

**Model Complexity Comparison:** Table 9 presents a comparison of model complexity across state-of-the-art manipulation localization networks. **BioTamperNet** achieves the best trade-off between computational cost and detection performance. While maintaining a high input resolution of $512 \times 512$, BioTamperNet reduces the model size to **36.7M** parameters and the computation to **29.6G FLOPs**, significantly lower than recent models like MVSS Chen et al. (2021), CAT-Net Kwon et al. (2021), and TruFor Guillaro et al. (2023), which require more than twice the computational cost. Notably, BioTamperNet is also more efficient than SparseViT Su et al. (2025), achieving a **41% reduction in FLOPs** and smaller parameter count, despite delivering better or comparable detection accuracy. These results highlight BioTamperNet's scalability and suitability for both high-fidelity and real-time forensic applications.