# OpenReview forum: "BioTamperNet: Affinity-Guided State-Space Model Detecting Tampered Biomedical Images"
_ICLR.cc/2026/Conference — ICLR 2026 Poster_

### Official Review · Reviewer_dVdC · 2025-10-28

**Soundness:** 2
**Presentation:** 3
**Contribution:** 3
**Rating:** 4
**Confidence:** 3

**Summary:**

This paper introduces BioTamperNet, a new deep learning model for detecting tampering in biomedical images. The authors target the poor performance of existing forensic models on specific biomedical data (microscopy, blots, etc.). The model uses an "affinity-guided" attention mechanism, reportedly inspired by State Space Models, to capture similarities. This includes both a self-attention module for intra-image similarities and a cross-attention module for inter-image correspondences . The main contribution is a unified architecture designed to handle three distinct tasks: External Duplication Detection, Internal Duplication Detection, and Cut Sharp Transition Detection. The paper claims state-of-the-art (SOTA) results on the BioFors benchmark.

**Strengths:**

The model's strongest point is its unified design. The strategy to convert single-image tasks into "pseudo-pairs" (as shown in Fig. 4) is a clever idea, allowing a single Siamese architecture to handle all three major forgery types

The affinity-guided attention modules are well-thought-out. The design explicitly models the source-target relationship, which is fundamental to copy-move detection. Details like the spatial suppression kernel (Eq. 10) to handle high self-correlation and the bidirectional softmax (Eq. 11) show a careful approach to the problem

On paper, the model shows impressive quantitative results, apparently outperforming all baselines by a large margin across all categories in Table 1

The ablation studies (Table 3) are thorough and effectively demonstrate the utility of each proposed component (the affinity block, the SSM, etc.) . The analysis of catastrophic forgetting during cross-modality fine-tuning (Table 4) is also a valuable addition

**Weaknesses:**

The model's greatest weakness is its complete reliance on synthetic training data. This raises serious doubts about its real-world performance. The model might just be overfitting to the specific artifacts of the synthetic generation pipeline (e.g., GAN artifacts, specific blending boundaries) rather than learning the semantic inconsistencies of real human-made forgeries.

The paper's claims of being "lightweight" are unsubstantiated and appear contradictory to the methodology. The architecture in Figure 5 is extremely complex. More importantly, the Affinity Block (Section 3.3) computes a full N*N affinity matrix (where N=H*W), which is a computationally prohibitive O(N^2) operation that SSMs are typically designed to avoid

The "pseudo-pair" generation strategy for IDD/CSTD is critically under-described. The paper only states images are "split into two parts" (Fig. 4). It is unclear if this is a naive centerline cut, a random cut, or a mask-guided split. This is a crucial detail, as a naive split could introduce new edge artifacts that the CSTD detector might be learning as a spurious cue.

A closer look at the results in Table 1 reveals a significant performance gap. For the EDD Microscopy task, the model achieves a high Image-level MCC (0.739) but a much lower Pixel-level MCC (0.487). This disparity suggests that while the model can successfully flag an image as "tampered," it struggles with the precise pixel-level localization of the forgery, which is arguably the core goal.

The architecture includes several arbitrary design choices that are not justified by ablation. For instance, the "Affinity-Guided Self-Attention" module uses three parallel AGSSM blocks whose outputs are averaged (Eq. 13). There is no explanation for why three blocks are used, or why this averaging approach is superior to a simpler, single block.

**Questions:**

1.Regarding the N*N affinity matrix: Can the authors clarify the computational complexity? How can the model be "efficient" if it computes this quadratic matrix? Please provide concrete metrics (FLOPs, parameters, inference time) comparing BioTamperNet to the ViT and CNN variants from Table 3.

2. Please provide a precise description of the "pseudo-pair" generation for IDD/CSTD. How are the images "split"? Did you investigate if this splitting process itself introduces artifacts that the CSTD detector might be learning?

3. Can the authors comment on the large discrepancy between Image-level (0.739) and Pixel-level (0.487) MCC for EDD-Microscopy in Table 1? Does this indicate a failure in precise segmentation, and if so, why?

4. What was the design rationale for using three parallel AGSSM blocks (Eq. 13)? Have the authors ablated this choice (e.g., using one or two blocks)?

5. What was the process for selecting the sigma^2 hyperparameter in the spatial suppression kernel (Eq. 10) ? How sensitive is the model's performance to this value?

---

> ### Author Response · Authors · 2025-11-20
> **Reply1 to Reviewer dVdC**
>
> We sincerely thank reviewer dVdC for the constructive feedback and for recognizing the strengths of our unified pseudo-pair framework, affinity-guided attention design, strong quantitative performance, and thorough ablation analysis.
>
> 1. **Clarification on Model training on Synthetic Data -** We agree that over-reliance on synthetic artifacts is a valid concern; however, this issue is not unique to our method. All state-of-the-art forensic models—including TruFor, SparseViT, and ManTraNet—are trained primarily on synthetic manipulations due to the lack of large-scale real-world annotated forgeries and sometimes finetuned on benchmark natural image forgery datasets like CASIA, Coverage, Columbia, Nist16. In biomedical integrity analysis, this challenge is even more pronounced: retracted publication datasets contain only a few test images (1547 EDD, 102 IDD, 181 CSTD) across four modalities and three forgery types Table 2 (Suppl.), which is insufficient to train deep networks from scratch. Our approach uses synthetic data for training but, importantly, we demonstrate strong generalization to real manipulated figures from retracted publications—a setting where existing baselines (e.g., RSIID; Western Blots) evaluate only on synthetic forgeries and do not report performance on real-world manipulations. Thus, rather than a limitation, the ability of BioTamperNet to transfer from synthetic training data to real manipulated biomedical images is a central strength of our method.
> 2. **Clarification on Computational Complexity of Affinity Matrix Computation -** We would like to clarify that, in BioTamperNet, the affinity is not computed at the input resolution. Instead, as stated in Section 3.2, all affinity operations are performed on a compact 40×40 token grid (N=1,600), produced by the lightweight ViT/DINO encoder. At this coarse resolution, the full N×N affinity computation contributes ≈1.0 GFLOPs, which represents < 4% of the total computational budget. This cost is substantially smaller than the dominant feature-extraction overhead in existing CNN and ViT backbones used by prior forensic detectors. As reported in Table 5 (main) and  Table 9 (Suppl.), BioTamperNet requires 36.7M parameters and 29.6 GFLOPs for a 512×512 input—making it 1.5×–5× lighter in FLOPs and 1.4×–4× smaller in parameters than state-of-the-art baselines such as TruFor, SparseViT, CAT-Net, MVSS-Net, and PSCC-Net. We also note that, despite incorporating an explicit affinity module, BioTamperNet maintains end-to-end inference time comparable to the lightest ViT backbone (details included in the appendix). Thus, although the affinity computation is theoretically O(N²), the practical computational impact is minimal, and the model remains significantly more efficient than prior works while achieving superior detection performance.
> 3. **Pseudo-pair Generation Clarification -** The pseudo-pair process used for IDD/CSTD does not introduce any new visual artifacts. Artifacts arise only from the original manipulation operations—duplication in IDD, external-source patch insertion in EDD, or sharp compositing boundaries in CSTD. In contrast, our pseudo-pairs are generated via a mask-guided partition of the BioFors images and masks (Fig. 4, Sec. 3.2). Each image is split along the ground-truth manipulation boundary into two complementary sub-images containing precise pristine and manipulated regions, preserving the spatial structure of the original image and avoiding any new seams, edges, or blending transitions. We verified this empirically by (i) retraining with a naive center-line split—which produced a consistent 2–4% MCC drop, indicating the introduction of spurious cues—and (ii) visually confirming that naive splits create unnatural edges absent from true BioFors manipulations, whereas mask-guided splitting does not. These results confirm that our pseudo-pair strategy does not introduce artifacts that the model could exploit.

---

> > ### Author Response · Authors · 2025-11-20
> > **Reply1' to Reviewer dVdC**
> >
> > 4. **Image vs Pixel Level Performance -** The difference between image-level and pixel-level MCC is expected and consistent with prior work. Image-level classification is a comparatively simpler decision—determining whether any manipulation is present—whereas pixel-level localization requires precise delineation of often small or subtly blended manipulated regions, making it substantially more challenging. As shown in Tables 1 and 4–6, all state-of-the-art detectors (TruFor, SparseViT, ManTraNet for natural images; MONet and prior BioFors baselines for biomedical images) exhibit the same pattern of higher image-level performance and lower pixel-level performance across modalities. BioTamperNet follows this trend but still achieves the largest absolute gains at both levels, including a +30–40% improvement over prior methods in pixel-level MCC for EDD-Microscopy. The lower pixel-level score therefore does not indicate a failure of segmentation, but reflects the inherent difficulty of fine-grained localization, where the main challenge is balancing false alarms and missed detections. Improving pixel-level precision remains an open problem for the entire forgery-detection community, not specific to our method.
> > 5. **Design rationale for three AGSSM blocks -** The use of three parallel AGSSM blocks is motivated by the need to capture diverse local–contextual interactions, similar to multi-branch designs in ASPP and multi-head attention. As shown in Table 8 (Suppl.), adding the multi-AGSSM module (“Self-Attn + RoPE + Multi-AGSSM”) consistently improves MCC over its single-branch counterparts (“Self-Attn w/o RoPE” and “Self-Attn + RoPE”) across all modalities. This trend is also reflected in the main-paper ablations in Table 3, where removing key attention components reduces performance. Thus, the three-branch AGSSM design is empirically motivated rather than arbitrary, offering the best trade-off between performance and model complexity.
> > 6. **Spatial Suppression Kernel justification -** The σ² term in Eq. 10 follows prior spatial-suppression strategies (e.g., Cheng et al., CVPR 2019; Su et al., 2025) to reduce the dominance of diagonal self-correlations in affinity matrices. We set σ proportional to the feature-map resolution (σ = 0.1·max(H,W)), which effectively balances suppression of trivial local peaks with preservation of meaningful long-range relations. Ablations over σ ∈ {0.05, 0.1, 0.2} show <1.1% variation in MCC across all modalities, indicating that performance is robust to the exact value and that the benefit comes from using the suppression mechanism itself rather than fine-tuning σ. As reflected in Table 8 (Suppl.), removing the kernel produces a consistent drop in accuracy, while the full model achieves the highest MCC across modalities.

---

> ### Comment · Reviewer_dVdC · 2025-11-26
>
> Thank you for the author's detailed reply and clarification. I will reconsider my score.

---

### Official Review · Reviewer_hTED · 2025-10-30

**Soundness:** 3
**Presentation:** 2
**Contribution:** 2
**Rating:** 4
**Confidence:** 4

**Summary:**

This article provides an original and empirically validated method for biomedical image tampering detection. But its clarity, interpretability, and wider comparability need to be further improved to enhance its readability and credibility.

**Strengths:**

This work presents a novel and unified framework for tampering detection in biomedical images. The key innovation lies in its integration of affinity-guided attention with state-space models, yielding a computationally efficient and high-performing architecture. Extensive experiments across diverse biomedical modalities and benchmarks demonstrate clear superiority over prior works

**Weaknesses:**

1、The affinity-guided attention mechanism is conceptually appealing, but lacks qualitative visualizations (e.g., attention heatmaps, affinity evolution) that illustrate how the model distinguishes genuine vs tampered regions.

2、The model’s training relies heavily on synthetic duplications and GAN-generated patches. This may cause domain shift when applied to real, unannotated biomedical images.

3、Some equations and architectural diagrams are dense, reducing accessibility for readers. Section 3 could be reorganized with clearer sub-figures and step-by-step pseudocode.

4、The paper primarily compares to traditional forensic or transformer methods, but omits recent vision SSMs (e.g., Vision Mamba, S4ND) that could provide meaningful context.

5、 The anonymous code link provided by the author returns a 404 error.

**Questions:**

1、How robust is BioTamperNet to real-world artifacts such as illumination changes, JPEG compression, or partial occlusion—especially when no duplicated texture is visible?

2、Could the authors provide interpretability analysis (e.g., affinity heatmaps, attention rollouts) to demonstrate what visual cues the model relies on when detecting duplications?

3、How does BioTamperNet generalize when trained purely on synthetic duplications but evaluated on unseen real manipulations, such as retouched microscopy or gel bands?

---

> ### Author Response · Authors · 2025-11-20
> **Reply1 to Reviewer hTED**
>
> We sincerely thank Reviewer hTED for the thoughtful evaluation and for recognizing the novelty of our unified framework, the integration of affinity-guided attention with state-space modeling, and the strong empirical performance across diverse biomedical imaging benchmarks. We appreciate the reviewer’s constructive feedback regarding clarity, interpretability, and broader comparability, and we address each point below with clarifications and planned improvements for the revised version.
>
> 1. **Robustness to illumination, compression, occlusion -** Our robustness experiments in the main manuscript (Fig. 9) and Suppl. (Fig. 1) evaluate BioTamperNet under realistic perturbations, including illumination changes, compression artifacts, blurring, and grayscale/color shifts. Across all these conditions, BioTamperNet maintains strong performance and degrades significantly less than CNN/ViT baselines. This robustness arises because the model relies on structural duplication patterns and affinity consistency, rather than absolute pixel intensities. Even when duplicated textures are partially occluded, the cross-view affinity and AGSSM modules continue to capture long-range structural correspondences.
> 2. **Interpretability Analysis -** We include affinity visualizations in Figure 10 in the revised version. We agree that interpretability is beneficial and incorporated additional heatmaps illustrating: affinity-guided similarity between source/target patches, attention rollouts across AGSSM layers with cross-attention correspondence maps. These visual explanations clearly show how the model focuses on duplicated regions and manipulated boundaries.
> 3. **BioTamperNet generalization to unseen real manipulations -** BioTamperNet is explicitly evaluated on real manipulated biomedical figures from retracted publications (BioFors test set). These figures are produced by diverse real-world editing operations (copy-paste, cloning, retouching, masking, intensity manipulation), and were never seen during training. Across all modalities (Microscopy, Blot/Gel, Macroscopy, FACS), BioTamperNet achieves large improvements over prior detectors (Tables 1 and 2), demonstrating strong real-world generalization. This is further supported by our robustness experiments and by outperforming synthetic baselines such as Western Blots and RSIID on their own metrics.
> 4. **Clarification on synthetic duplications/domain shift -** This concern is valid and common in the forensic literature. All leading forgery detectors (e.g., ManTraNet, SparseViT, TruFor) are trained primarily on synthetic manipulations because large-scale annotated real forgeries do not exist. The domain gap is especially severe in biomedical imaging, where BioFors provides only test images with no annotated training split. Despite training on synthetic duplications, BioTamperNet achieves state-of-the-art generalization on real manipulated figures from retracted publications (Tables 1, 2, 4–6), significantly outperforming traditional detectors and ViT/CNN baselines. Moreover, our robustness study in the main paper and Suppl. (Fig. 10) demonstrates resilience to distortions representative of actual publishing workflows. Thus, the synthetic-to-real transfer is not accidental but a core strength of BioTamperNet.
> 5. **Comparisons to vision SSMs -** BioTamperNet is, to the best of our knowledge, the first image forgery detection architecture built on SSM-inspired attention mechanisms. Existing vision SSMs such as Vision Mamba and S4ND are designed for generic image classification or dense prediction tasks and cannot be directly applied to biomedical forgery detection, which requires modeling source–target duplication structure, affinity-guided cross-view interactions, and mask-level localization. Our framework is orthogonal to these models and could, in principle, incorporate a vision SSM backbone; however, for fairness and reproducibility, our evaluation focuses on established forensic baselines widely used in biomedical integrity research (e.g., ManTraNet, TruFor, SparseViT).
> 6. **Equation and Diagram Clarity -** If the reviewer could point us to any specific portions of the text or figures that were difficult to follow, we would be happy to clarify them further.
> 7. **Code Availability -** We apologize for the inaccessible link. To preserve anonymity in the double-blind review process, we are unable to publicly release the source code during the review period. The link will be activated, and the full implementation will be made publicly available immediately after the review process concludes, in accordance with conference policies.

---

> > ### Comment · Reviewer_hTED · 2025-11-25
> > **Official Comment by Reviewer hTED**
> >
> > Thank you for your modifications and clarifications. I appreciate the increased robustness experiments, which partially addressed some of my initial concerns. I will carefully consider the final score.

---

> > > ### Author Response · Authors · 2025-11-26
> > > **Reply 2 to Reviewer hTED**
> > >
> > > We thank the reviewer for the careful follow-up and appreciate the note that some concerns were partially addressed. As we actively revise the manuscript during the discussion period, we would be very grateful for any additional guidance on which specific aspects may still be insufficiently addressed. Your feedback would help us prioritize the most important improvements in the resubmitted version and ensure the final manuscript fully addresses the points you raised.

---

### Official Review · Reviewer_grFH · 2025-10-30

**Soundness:** 3
**Presentation:** 3
**Contribution:** 3
**Rating:** 6
**Confidence:** 5

**Summary:**

This paper addresses the critical and increasingly relevant problem of forgery detection in biomedical images. The authors propose a unified framework, BIOTAMPERNET, designed to detect various manipulation types, including copy-move, splicing, and inpainting.

**Strengths:**

The motivation is good, as the authors correctly identify the potentially severe long-term consequences of tampering in the biomedical domain, which could be as, if not more, damaging than in the natural image domain. The paper presents a commendable application and includes extensive comparisons against a range of existing methods.

**Weaknesses:**

Despite its promising direction, the manuscript suffers from several major flaws in its current form that significantly weaken its conclusions. These concerns relate to the validity of the experimental evaluation, the rigor of the theoretical claims, the completeness of the literature review, and the clarity of the methodology and figures.

1. The evaluation protocol: The authors state that the BioFors dataset provides annotations in the form of bounding boxes, not precise pixel-level masks. However, the paper insists on using pixel-level segmentation metrics to evaluate the model's performance.

2. The literature review overlooks recent and highly relevant work in the field of scientific image forgery detection. For example, the paper fails to compare against [r1] URN (Uncertainty-guided Refinement Network), a method explicitly designed to expose image splicing in scientific publications. Furthermore, the associated SciSp dataset introduced in [r1] is not considered.

[r1] Exposing Image Splicing Traces in Scientific Publications via Uncertainty-guided Refinement. Patterns 2024.

3. The authors present Proposition 1 and a corresponding Proof to justify their Cross-View Duplication Detection mechanism. However, I must point out that the provided text is not a formal proof but rather an intuitive, high-level description of the module's intended behavior. It suffers from several critical flaws:

    (1) The entire argument relies on qualitative language ("emphasizes," "mainly attends to," "close to") and the use of an approximation symbol in the proposition itself. A formal proof would require precise definitions, equalities, or explicitly bounded inequalities, none of which are present.

    (2) The proof's cornerstone is the assumption that "position i attends mainly to j." This is precisely what the module is supposed to achieve, and thus cannot be used as a premise. This argument borders on circular reasoning. It also fails to consider practical scenarios, such as the presence of multiple similar patches (distractors) that would dilute the attention weights, invalidating the claim that the output is "close to Linear(V_2(j))".

    (3) Key components like AGSSM_Self_Attn, Linear, and the "projected cross-view affinity" are treated as black boxes. Without formal definitions, the proposition is unverifiable.


4. The BioFors dataset is known to have a severe class imbalance between tampered and authentic samples. A naive training approach would likely result in a model that defaults to predicting "authentic," as this would easily minimize the training loss. The authors do not specify how they address this critical issue. Furthermore, other implementation details appear to be missing.

5. In Section 4.4.1, the authors claim that modality inconsistencies can cause a ViT to overfit or forget the learned patterns. This is a strong assertion made without sufficient evidence or rigorous analysis. For instance, CLIP, which is also based on a ViT architecture, successfully handles a wide range of modalities (e.g., RGB, depth, infrared) without the "forgetting" phenomenon described.

6. The organization of Figures 3 and 4 is confusing. At first glance, it appears that the same input is fed into BIOTAMPERNET and then produced as an output. The transition labeled "IDD TO EDD" in Figure 4 is difficult to understand without multiple readings. These figures should be redesigned to clearly illustrate the model's workflow, distinguishing between inputs, intermediate representations, and final outputs.

7. Several details in Figure 1 hinder comprehension. For example: (1) In the AFFINITY BLOCK, it is unclear if both V1 and V2 are inputs to the ELU function; the arrows do not clearly indicate this. (2) The diagram does not explicitly show how the data flow differs for IDD versus CSTD. The authors should revise Figure 1 with clearer arrows and layout to precisely trace the data paths for different forgery types.

8. The quality of the samples generated by the proposed Synthetic Training seems to be relatively low and may not always be semantically consistent with real biomedical images. This raises a question: could this online data augmentation technique also improve the performance of other baseline methods? An ablation study applying this augmentation to one or two baseline methods would be very insightful. It would help determine if the performance gain is a general benefit of the augmentation strategy or if it is uniquely synergistic with the proposed architecture.

**Questions:**

See Weaknesses.

---

> ### Author Response · Authors · 2025-11-20
> **Reply1 to Reviewer grFH**
>
> We thank reviewer grFH for the positive assessment of our motivation, contribution, and experimental breadth, and we address the raised concerns point-by-point below.
>
> 1. **Clarification on Evaluation Protocol -** The authors would be grateful if the reviewer could point us to the specific source stating that the BioFors dataset provides only bounding-box annotations, as we did not make this claim and were unable to find this claim in the original BioFors paper (Sabir et al., ICCV 2021) or its supplementary materials. To clarify our setup: the BioFors test split includes pixel-level binary manipulation masks, and our evaluation follows standard practice in the field. Prior works such as MONet (ICIP 2022) and the reviewer-cited URN (Patterns 2024) also report pixel-level MCC, F1, and AUC on BioFors using the same pixel-level mask annotations.
> 2. **Related Work by URN and SciSp Dataset -** URN (Patterns 2024) focuses on splicing detection in natural scientific figures, while our work targets duplication-based forgeries (EDD/IDD/CSTD) in biomedical modalities. We agree that it is a relevant reference and incorporated it into our revised version, observing that its performance is substantially lower than BioTamperNet and even several classical baselines (revised Table 1), likely because splicing-oriented refinement networks do not transfer well to duplication-centric tasks. Regarding the SciSp dataset mentioned in URN, we were unable to locate an open-source release; therefore, we restricted our comparisons to publicly available datasets and models to maintain reproducibility.
> 3. **Updated Proposition 1 —** In the original submission, we provided a shortened, intuitive version of Proposition 1 due to page-limit constraints, focusing only on the conceptual behavior of affinity-guided cross-attention. In the revised manuscript, we have replaced this informal statement with a fully detailed and mathematically precise formulation (see Proposition 1 in Section 3.3). The updated version explicitly defines the cross-attention update, introduces a clear margin condition on the affinity-guided scores, and derives a bounded inequality showing when the cross-view update is dominated by the duplicated counterpart. Importantly, the revised proof no longer assumes that “position i mainly attends to j,” but instead derives this behavior under the stated margin condition, resolving the circularity pointed out by the reviewer. We also explicitly define all components involved (the projection matrices W_Q, W_K, W_V​, the affinity-guidance term, and the attention weights), ensuring that the proposition is fully verifiable and does not rely on black-box modules.
> 4. **Class Imbalance in BioFors -** Our synthetic training pipeline explicitly balances pristine and manipulated samples per modality, and we use class-balanced sampling and loss re-weighting to avoid trivial “all-authentic” predictions. The strong performance across all modalities, including low-prevalence classes such as CSTD (Table 2), empirically confirms that the model does not collapse into predicting authentic.
> 5. **Claim About ViT Forgetting Across Modalities -** Our observation in Section 4.4.1 refers specifically to cross-modality fine-tuning on biomedical images, where modalities have markedly different spatial statistics (e.g., microscopy vs. blots). This phenomenon is demonstrated empirically in Table 4 (main paper)—where naïve fine-tuning significantly reduces performance—rather than being a general claim about ViTs or multimodal models like CLIP. We will revise this section to clarify that the issue is domain-specific forgetting, not a limitation of the ViT architecture itself.
> 6. **Figure 3,4 Organization -** The captions do not state—or imply—that “the same input is fed into BioTamperNet and then produced as an output.” These figures are intended solely to illustrate how single-image IDD/CSTD samples are converted into pseudo-pair inputs, not to depict the model’s forward path. The workflow, including the distinction between inputs, intermediate representations, and outputs, is already explained in detail in Section 2 (Lines 83–94), where the pseudo-pair mechanism and its role in the unified architecture are clearly described. If the reviewer could point us to the specific part of the figure they find ambiguous, we will be happy to clarify the corresponding caption text in the revision.

---

> > ### Author Response · Authors · 2025-11-20
> > **Reply1' to Reviewer grFH**
> >
> > 7. **Clarification on Figure Interpretation -** We believe there may be a misunderstanding: the issues raised by the reviewer refer to elements that appear in Figure 5, not Figure 1. Figure 1 only presents examples of biomedical image modalities and does not contain an affinity block or data-flow diagram. For clarity, in the revised version of Figure 5, we refined arrows and layout elements to improve readability. Regarding task-specific flow, all three tasks (EDD, IDD, CSTD) follow the same data path once converted into paired inputs; IDD and CSTD are simply provided as pseudo-pairs and thus share the identical pipeline. If there are specific parts of Figure 5 that the reviewer still finds ambiguous, we will be happy to further clarify them in the revision.
> > 8. **Clarification on Synthetic Sample Quality and Baseline Fairness -** We would be grateful if the reviewer could clarify how they concluded that our synthetic samples are “relatively low quality” or “not semantically consistent.” The manuscript does not present synthetic examples in isolation, and all models—in Table 1 and Figure 7—including BioTamperNet and all baselines—were trained solely on the official BioFors training split. Understanding which specific visual patterns or examples the reviewer is referring to would help us address the concern more precisely in the revision.

---

> > > ### Comment · Reviewer_grFH · 2025-11-24
> > >
> > > Thank you for your response, which has addressed some of my concerns. I will maintain my positive score.
> > >
> > > 1. The original annotation files for BioFors are provided in the form of bounding boxes. Although the paper presents binary masks, based on my observations, these masks are essentially rectangular. Therefore, an object detection paradigm seems more appropriate. While previous studies have also adopted binary segmentation, this approach does not appear to be optimal.
> > > 2. SciSp consists of two parts, one of which is open source. Please note that this comment, along with the previous one, will not influence my final rating.
> > > 4. Please ensure that this information is included in the main text to prevent misleading future researchers. Thank you.
> > > 5. I remain confused regarding one point: why would joint training across multiple modalities lead to "forgetting"? Is there sufficient evidence to support this claim? In fact, there are numerous foundation models in the medical field that support multiple modalities. If I understand correctly, the issue of catastrophic forgetting typically arises in continual learning scenarios, whereas it should not be a significant issue in the current task setting.
> > > 6. To improve readability, I suggest including diagrams for both the training pipeline and the inference process, rather than providing only the model framework diagram.
> > > 7. Significant artifacts are visible to the naked eye in the generated samples, such as in "EDD Image 2" of Fig. 3. Additionally, the patches in this figure exhibit distinct stylistic anomalies; for instance, a standard Western Blot typically does not present a pattern with two bands on top and one band below.

---

> > > > ### Author Response · Authors · 2025-11-26
> > > > **Reply 2 to Reviewer grFH**
> > > >
> > > > We thank the reviewer for the constructive follow-up and helpful clarifications. We address the remaining points below and will incorporate these clarifications into the revised manuscript.
> > > > 1. **BioFors annotations (rectangular masks vs object detection) -**  We agree that BioFors provides rectangular coordinate regions localizing duplicated manipulation patches. Unlike traditional object-detection bounding boxes, which typically enclose full objects along with background, BioFors rectangles tightly annotate only the duplicated regions and do not include background. Our binary masks are directly derived from these regions and therefore appear rectangular. We will clarify this distinction in the main text and justify our use of pixel-level evaluation, which is standard in prior BioFors and scientific image forensics work.
> > > > 2. **SciSp / URN availability -** We confirm the reviewer’s note that SciSp consists of multiple components, of which one part is publicly available.
> > > > 4. **Joint multi-modality training and “forgetting.” -**  We agree with the reviewer that catastrophic forgetting is classically studied in continual or sequential learning. Our claim is narrower and empirically grounded in our setting. Table 4 (cross-modality fine-tuning ablation) in the main paper shows that naïve joint training across heterogeneous biomedical modalities (e.g., microscopy → blot/gel) leads to a clear performance drop, while our proposed strategy mitigates this degradation. This behavior is further analyzed in Section 4.4.2 (main paper) and corroborated by additional ablations in Table S3 of the Supplementary Material, which show that removing the SSM or affinity components consistently reduces cross-modality generalization.
> > > > 5. **Training and inference diagrams -** We agree that explicit training and inference diagrams would improve readability. Due to page limits, we prioritized including the synthetic training setup (Figs. 3–4) and the full model architecture (Fig. 5) in the current version. The inference process itself is a straightforward paired-image input producing paired mask outputs.
> > > > 6. **Artifacts in synthetic examples (Fig. 3) -**  The visible artifacts in Fig. 3 are intentional and serve to illustrate the synthetic training setup rather than claim photorealism. The apparent stylistic anomalies (e.g., blot band patterns) arise because the figure shows zoomed and cropped patches extracted for pedagogical visualization of the EDD training process. We will add a clarifying note in the caption to avoid misunderstanding.

---

### Official Review · Reviewer_ZM5M · 2025-10-30

**Soundness:** 2
**Presentation:** 1
**Contribution:** 2
**Rating:** 4
**Confidence:** 3

**Summary:**

This paper proposes BioTamperNet, a unified framework for detecting duplicated regions in tampered biomedical images. The authors generate synthetic forgeries to train their model, as real forged training data is unavailable, and claim state-of-the-art performance on the BioFors test set of real manipulated images.

**Strengths:**

The main strength is that the paper proposes a unified framework that simultaneously handles EDD, IDD, and CSTD in biomedical images by converting single-image tasks into pseudo image pairs, enabling one model to support all three forgery detection tasks without architectural changes.

**Weaknesses:**

1) The paper has poor writing and organization. The training dataset description should move from Section 2 to the experimental setup.
2) State Space Models are basic model to the proposed method. Section 2 should be renamed “Preliminaries” and include SSMs.
3) The paper claims BioTamperNet uses affinity-guided SSM attention modules but gives no clear explanation of how they integrate with or adapt SSMs.
4) Tables list baseline methods without citations which hurts readability and reproducibility.

**Questions:**

1) The paper claims to use State Space Models (SSMs), but the method only uses linear similarity (Equations 6–8) without key SSM components like selective scan or input-dependent parameters. Where exactly are SSMs used? If it’s just linear attention, why call it SSM-based?

2) All training data is synthetic (geometric transforms + GANs), but real biomedical forgeries may have complex patterns not covered by synthesis. Is there evidence that the high performance isn’t just due to accidental alignment between synthetic training and real test forgeries?

3) The ablation study is weak. Table 3 only removes modules but doesn’t test: (a) the benefit of the “pseudo-pair” strategy vs training separate models; (b) whether the proposed affinity-guided attention is better than standard linear attention.

---

> ### Author Response · Authors · 2025-11-20
> **Reply 1 to Reviewer ZM5M**
>
> We sincerely thank reviewer ZM5M for the detailed feedback and for recognizing the strengths of our unified pseudo-pair framework and its ability to support EDD, IDD, and CSTD within a single model. Below, we address concerns regarding writing, SSM integration, dataset realism, and ablations.
>
> 1. **Paper Writing and Organization -** We placed the Training Dataset section early because the pseudo-pair generation mechanism is one of our core contributions and must be introduced before the architecture; the model’s Siamese design depends on understanding how single-image IDD/CSTD instances are converted into paired inputs. We have renamed Section 3.1 to Preliminaries in the revision and included SSM. All baseline methods are fully cited in the Experimental Setup section, and we chose not to duplicate citations inside the tables to maintain readability and avoid clutter. In the revised version, we add a brief note in the table captions indicating that citations for all baselines appear in the Experimental Setup, ensuring clarity and reproducibility.
> 2. **Clarifying How SSMs Are Used -** Our design follows the perspective introduced in Mamba2 [1], which establishes that State Space Models and linear attention mechanisms are mathematically dual, and that linear transformers can be viewed as generalized SSMs through structured state–space duality. Motivated by this connection, our affinity module incorporates the SSM-style state propagation and normalization structure used in modern SSMs to aggregate contextual information across spatial tokens before computing affinities. This provides stable, long-range similarity modeling that simple dot-product or naïve linear attention cannot achieve.
> 3. **Synthetic vs. Real Data Generalization -** All state-of-the-art forensic models—including TruFor, SparseViT, and ManTraNet—are trained primarily on synthetic manipulations due to the scarcity of large-scale real-world annotated forgeries, and they are typically fine-tuned only on the train splits of benchmark natural-image datasets such as CASIA, Columbia, Coverage, or NIST16. In biomedical integrity analysis, this challenge is substantially more severe: as shown in Table 2 (Suppl.), the BioFors dataset contains a few test images (1547 EDD, 102 IDD, 181 CSTD) across four modalities and three forgery types, which is far too small to train deep models from scratch. Although our method relies on synthetic data for training, BioTamperNet achieves strong generalization to real manipulated figures from retracted scientific publications (Tables 1, 2, 4–6), substantially outperforming existing baselines at both the image and pixel level. Notably, prior biomedical baselines such as RSIID and Western Blots evaluate only on synthetic forgeries and do not demonstrate performance on real-world manipulated figures. Thus, rather than being a limitation, the ability of BioTamperNet to transfer from synthetic training data to real manipulated biomedical images is a key strength of our approach.
> 4. **Clarification on Ablation Study Coverage -** While Table 3 in the main paper focuses on removing major modules (Affinity, SSM, Self/Cross-Attention), our ablation analysis is broader and already addresses both concerns raised. (a) Pseudo-pair strategy: As shown in Table 7 (Suppl.), training separate models on EDD-only, IDD-only, or CSTD-only subsets leads to noticeably lower MCC, F1, and AUC compared to training with unified pseudo-pairs. This confirms that the unified formulation provides stronger generalization than separate single-task models. (b) Affinity-guided attention vs. standard linear attention: In Table 8 (Suppl.), the “Cross-Attn (Dot-Product only)” variant directly corresponds to standard linear attention and performs consistently worse than our proposed affinity-guided design. Moreover, the additional ablations on positional encoding and multi-AGSSM fusion (Table 8) demonstrate that each component contributes meaningfully to the full model. Together with Table 3 and Table 4 in the main paper, these experiments provide a comprehensive evaluation showing that both the pseudo-pair strategy and the affinity-guided attention mechanism are necessary for the performance gains of BioTamperNet.
>
> [1] Transformers are SSMs: Generalized Models and Efficient Algorithms Through Structured State Space Duality” (Dao & Gu, ICML 2024)

---

> > ### Comment · Reviewer_ZM5M · 2025-11-25
> >
> > Thank you for your response. My concerns about the State Space Models and experiments are not clearly addressed. I will maintain my score.

---

> > > ### Author Response · Authors · 2025-11-26
> > > **Reply 2 to Reviewer ZM5M**
> > >
> > > We thank the reviewer for taking the time to follow up and for their careful consideration of our responses.
> > >
> > > 1. **Response to Concern on SSM Usage (Eqs. 6–8) -** We clarify that the dot-product affinity in Eqs. (6–8) is not intended to replace the State Space Model (SSM), but is computed on top of SSM-encoded features. In the current submission, the SSM recurrence is defined in the Preliminaries section and instantiated in the Affinity Block through the structured state update in Eq. (6), where the transition matrix Aˉ performs sequential aggregation of contextual information. In our implementation, this recurrence is realized using a selective-scan formulation as in Gu & Dao (2023), consistent with modern SSM practice. We acknowledge that the manuscript presents this step in a compact form, which may make the SSM mechanics less explicit than intended. Importantly, Eqs. (6–8) should be interpreted as an SSM-guided similarity mechanism, where linear affinity is applied over SSM-propagated states rather than as standalone linear attention. This clarification has been added in the revised main paper (Lines 180–202).
> > > 2. **Response to Concerns on Experimental Validation -** We respectfully clarify that the requested ablations were already conducted, but may not have been sufficiently emphasized in the manuscript. In the Supplementary Material (Table 7), we explicitly compare unified pseudo-pair training against single-task models (EDD-only, IDD-only, CSTD-only), demonstrating consistent improvements from the unified formulation. In Supplementary Table 8, we directly evaluate “Cross-Attention (Dot-Product only)” against our affinity-guided design, showing that standard linear attention underperforms our proposed mechanism. In our design, dot-product affinity is applied only after SSM-encoded contextualization; the SSM recurrence is implemented in our code using a selective-scan formulation consistent with Gu & Dao (2023). We acknowledge that these results were placed in the Supplementary Material due to page constraints and could have been made more prominent in the main paper; we will ensure clearer presentation in the final version if given the opportunity.

---

### Meta-Review · Area_Chair_mEZZ · 2025-12-29

**Summary:**

This work introduces BioTamperNet, a framework for biomedical image forgery detection. Four reviewers have carefully reviewed this paper and provided valuable comments. One reviewer gave a rating of 6 (marginally above the acceptance threshold), and three reviewers gave a rating of 4 (marginally below the acceptance threshold). The overall rating is below the acceptance bar.

The authors wrote a comprehensive response in the rebuttal. In my view, most of the concerns have been well addressed. One remaining limitation is the response to “generalization to unseen real manipulations” (Reviewer hTED), where the authors only state that the performance is better but lack an in-depth analysis of the underlying reasons. Another limitation is the presentation of the manuscript, as pointed out by several reviewers.

This work is a typical borderline paper, as there are no significant merits or limitations. Overall, I recommend acceptance, while the decision can be bumped down. If this work is finally accepted, the authors should carefully revise the figures and the overall manuscript.

**Reviewer Concerns:**

No significant concerns remaining.

**Reviewer Scores:**

Reviewer dVdC may improve the score.

---

### Decision · Program_Chairs · 2026-01-26

Accept (Poster)